# Copper integrative catalytic pairs with mixed-valence Cu²⁺-Cu³⁺ Species for selective alkyne conversion

Yuxue Yue [1], Mingde Yu[1], Zhangyi Yao[2], Guangzong Fang [3], Bolin Wang[4], Saisai Wang[1], Chunxiao Jin[1], Renqin Chang[5], Tulai Sun[1], Zhiyan Pan[6], Yihan Zhu [1,5], Feng Ryan Wang [2] ✉, Xiaonian Li [1] ✉ & Jia Zhao [1] ✉

Achieving specific orbital activation of $C \equiv C$ by controlling the precise atomic architecture of supported metals is crucial for the selective transformation of alkynes. However, its physical mechanism remains a subject of debate. Herein, we construct a well-defined O-bridged $CuN_3$-O-$CuN_3$ integrative catalytic pairs (Cu ICPs) based on Kirkendall effect. As a result, Cu ICPs with mixed $Cu^{2+}$-$Cu^{3+}$ species demonstrate >99% conversion and >550 h stability in acetylene hydrochlorination (simulated industrial reaction conditions), showcasing unparalleled performance in the liquid-phase hydrochlorination of five alkynes as well. A combined experimental and theoretical analyses reveal selective coupling between the $d_{xz}/d_{yz}$ orbitals of Cu ICPs and the $\sigma$ orbitals of $C \equiv C$ in $C_2H_2$, leading to the formation of highly reactive di-$\sigma$-HC = CH intermediate. Additionally, the presence of the bridged-O species promotes HCl dissociation, altering the addition pathway from the classical Eley-Rideal (E-R) mechanism to a Cl•-trigged Langmuir-Hinshelwood (L-H) mechanism, ultimately reducing the intrinsic energy barrier for addition, and breaking the universal standard electrode potential linear scaling relations.

Catalytic performance, characterized by activity, selectivity, and stability, is fundamentally linked to the atomic structure of active sites, determining the efficiency and pathway of targeted chemical reactions[1–4]. Recent advancements in controlling the atomic configuration of supported metal particles—down to the single-atom scale—have propelled dual-atom catalysts (DACs) even multi-atom site catalysts to the forefront of nanoscience[2,5–9]. DACs, with their synergistic interactions between paired metal atoms, possess a unique ability to activate challenging chemical bonds in complex compounds, often outperforming traditional nanoparticle and single-atom catalysts (SACs)[1,3,10]. However, defining the precise nature of diatomic sites remains a challenge due to the diverse interactions between paired atoms, whether they act as active centers, coordination ligands, or long-range structural modifiers.

One such enduring challenge is the efficient catalytic transformation of alkynes, a cornerstone in both scientific research and industrial applications, given its crucial role in synthesizing essential commodities[1,11–13]. A relevant example is acetylene hydrochlorination, which has garnered renewed attention due to the urgent need to replace toxic mercuric chloride catalysts traditionally used in producing vinyl chloride monomers (VCM)[14–17]. This production process, which underpins the extensive demand for polyvinyl chloride in construction and piping, currently consumes over 1200 tons of mercury annually, with substantial environmental repercussions: more than 550

[1]Zhejiang Key Laboratory of Surface and Interface Science and Engineering for Catalysts, Institute of Industrial Catalysis of Zhejiang University of Technology, Hangzhou, China. [2]Department of Chemical Engineering, University College London, London, UK. [3]State Key Laboratory of Catalysis, Dalian Institute of Chemical Physics Chinese Academy of Science, Dalian, China. [4]School of Chemical Engineering, Northeast Electric Power University, Jilin, China. [5]Research Center of Analysis Measurement, Zhejiang University of Technology Hangzhou, Hangzhou, China. [6]College of Environment, Zhejiang University of Technology, Hangzhou, China. ✉e-mail: ryan.wang@ucl.ac.uk; xnli@zjut.edu.cn; jiazhao@zjut.edu.cn

tons of mercury are wasted, and 40 tons are released as vapor into the atmosphere[18,19]. The urgency to develop environmentally friendly alternatives aligns with the global goals of the Minamata Convention. Noble metal catalysts (e.g., Au, Pd, Ru, Pt, Ir) have shown potential as mercury-free replacements[14,19–25]; notably, Au-based catalysts have already been commercialized by Hutchings et al. for this process in 2015[14,19], with a production facility currently operating in Shanghai, China. Recently, copper-based SACs have attracted attention due to their favorable cost-performance ratio and the tunable structure of their active sites[26–28]. Yet, issues like metal aggregation at high loadings, coke deposition, and weak acetylene adsorption limit their long-term stability and catalytic efficiency. This is mainly attributed to the degree of filling of the antibonding states of Cu on adsorption, and the degree of orbital overlap with the adsorbate.

Inspired by recent advances in the versatile catalytic properties of DACs[1,3,10], this work introduces a well-defined catalyst featuring integrative copper catalytic pairs (Cu ICPs), designed to address the limitations of conventional Cu SACs. These Cu ICPs are composed of two interconnected $CuN_3$ units, each serving as an active center within the catalytic cycle, forming a $CuN_3$-O-$CuN_3$ integrative pairs that enable cooperative site-to-site electronic interactions. Experimental and computational mechanistic study revealed that the Cu ICPs generate mixed-valence copper ($Cu^{2+}$-$Cu^{3+}$) species, with $Cu^{3+}$ demonstrated to exhibit exceptionally high catalytic activity. The coupling sites of Cu ICPs modify the conventional $\pi$-adsorption mode of acetylene, commonly observed on single-atom sites, into a more reactive di-$\sigma$ adsorption configuration. Meanwhile, bridged O atoms in Cu ICPs promote efficient HCl activation and dissociation, accelerating the overall reaction. The resultant Langmuir-Hinshelwood (L-H) mechanism on the Cu ICPs, involving the formation of the *CH=CHCl intermediate, facilitates a lower energy barrier via an unconventional rate-determining step, leading to superior activity compared to the Eley-Rideal (E-R) mechanism on CuCl, $CuCl_2$, and $CuN_4$ sites. By highlighting the structural and compositional strengths of Cu ICPs, this study opens new avenues for the design of precise and efficient catalysts, particularly for selective alkyne addition reactions. The findings underscore the potential of ICPs to transcend traditional catalytic limitations, offering a blueprint for future innovations in sustainable chemical processes.

## Results

### Synthesis and characterization of Cu ICPs

The synergetic two-step pyrolysis and coordination-etching approach[29,30] introduced here comprises a versatile and scalable method for the preparation of Cu ICPs catalyst, primarily by controlling metal ion migration and coordination induction to construct integrative catalytic sites. Figure 1a illustrates the sequential formation process of the Cu ICPs catalyst Cu-NOC. This process begins with the preparation of copper phthalocyanine (CuPc) nanorods, which serve as templates for the subsequent coating procedure. The binary mixture of ionic liquids (ILs) [BMIm]Cl and [EMIm]NTf2 (co-ILs) is combined with CuPc nanorods to form CuPc@co-ILs core-shell capsule-like composites. The resulting composite is then heated to 673 K under $N_2$ atmosphere for pre-polymerization[31]. Following this, a further pyrolysis assisted with coordination etching is used to synthesize the Cu ICPs. Specifically, the pre-polymerized sample is further ultrasonically treated in a mixture of hydrogen peroxide and hydrochloric acid (1:3 $H_2O_2$:HCl)[32] for 3 h, then calcined at 973 K for 2 h in $N_2$ to control metal migration and O-bridge coordination, ultimately yielding the prepared Cu ICPs catalyst (Cu-NOC-973). With this method, one is able to synthesize the Cu-NOC-973 with a high amount of >60 g/batch at the laboratory scale. It is noteworthy that the disappearance of the CuPc rods is not due to decomposition but to migration at high temperatures, known as the Kirkendall effect[33]. The formation of CuPc rods preventing the collapse and rupture of the shell as the hollow structure

develops. In addition, HR-TEM images in Supplementary Fig. 1 show that the average hollow pore diameter gradually increases with higher annealing temperatures, while the wall thickness decreases, which would offer a larger margin for Cu dispersion.

The morphologies of Cu-NOC-973 are characterized by scanning electron microscopy (SEM) and high-resolution transmission electron microscopy (HR-TEM). SEM images (Fig. 1b) show that the Cu-NOC-973 sample exhibits a nanocapsule morphology. Cross-sections obtained by fixing nanocapsule slices reveal an internal hollow channel. HR-TEM images (Fig. 1c and Supplementary Fig. 1d) also confirm the uniform hollow morphology of Cu-NOC-973. Energy-dispersive X-ray spectroscopy (EDS) maps (Fig. 1d) show that the elements Cu, C, N, and O are uniformly distributed throughout the hollow nanocapsules, and indicated that the Cu concentration in Cu-NOC-973 is 0.12 wt%, which is consistent with ICP-OES results (Supplementary Table 1). $N_2$ adsorption-desorption isotherms of the Cu-NOC-973 exhibit a typical type-IV isotherm with a distinct hysteresis loop (Supplementary Fig. 2), revealing a mesoporous structural characteristic. The Brunauer-Emmett-Teller (BET) surface area of Cu-NOC-973 is calculated to be 502.35 $m^2 g^{-1}$. The pore size distribution curve (Supplementary Fig. 3) derived from the Barrett-Joyner-Halenda (BJH) model suggests that Cu-NOC-973 possesses a uniform pore size of 3.05 nm (Supplementary Table 2), which ensures full contact between reactant molecules and Cu active sites and facilitates the mass transfer process during reactions.

To further validate the generality of synthetic strategy and optimize the coordination structure, several control experiments were conducted using the experimental protocols listed in Supplementary Fig. 4. First, a comparison sample, NOC-973, was synthesized using the same preparation procedure as Cu-NOC-973 but without the addition of the CuPc template. As expected, the NOC-973 exhibited a significantly irregular morphology (Supplementary Fig. 1a). Subsequently, hollow carbon nanocapsules formed at different calcination temperatures (673, 773, and 1173 K) in the presence of the CuPc template were also characterized using HR-TEM to track the evolution of the hollow structure. Supplementary Fig. 1b–e shows that the hollow nanocapsule structure of Cu-NOC forms only at synthesis temperatures above 773 K. Supplementary Figs. 5–10 provide corresponding EDS images, illustrating the morphology and spatial distribution of elements, which is consistent with the findings of Cu-NOC-973. Finally, the effect of different CuPc amounts on the morphology of the synthesized materials was systematically studied. As shown in Supplementary Fig. 4b; the Cu contents could be well controlled in Cu-NOC synthesized with designed Cu loading. The morphologies of 0.01Cu-NOC, 0.05Cu-NOC, and 0.2Cu-NOC are shown in Supplementary Fig. 11. All materials exhibit morphologies comparable to Cu-NOC-973 and possess a hollow nanocapsule structure, demonstrating that CuPc induces the formation of hollow carbon nanocapsules.

Additionally, atomic dispersion of Cu is directly observed and confirmed by aberration-corrected HAADF-STEM (HAADF-STEM) analysis (Fig. 1e and Supplementary Fig. 12), which shows a large concentration of paired bright dots (marked with white circles) coupled with a small amount of individual bright dots (marked with red circles), corresponding to integrative Cu pairs and single atoms, respectively. The statistical analysis of >20 dual-atom pairs revealed that the integrative Cu pairs account for 90% of all discernible bright spots, while the distance between neighbouring atoms is ~0.37 nm (Supplementary Fig. 12), which is higher to the Cu-Cu bond length (0.23–0.25 nm) in bulk Cu, indicating that the Cu atoms are not directly bonded to each other. Figure 1f, g shows the magnified corresponding three-dimensional (3D) atom-overlapping Gaussian-function fitting mappings of the regions marked by yellow cycles in Fig. 1e, well demonstrating the pairing feature for Cu ICPs in Cu-NOC-973 catalyst. Specifically, the distance between a pair of adjacent Cu atoms was

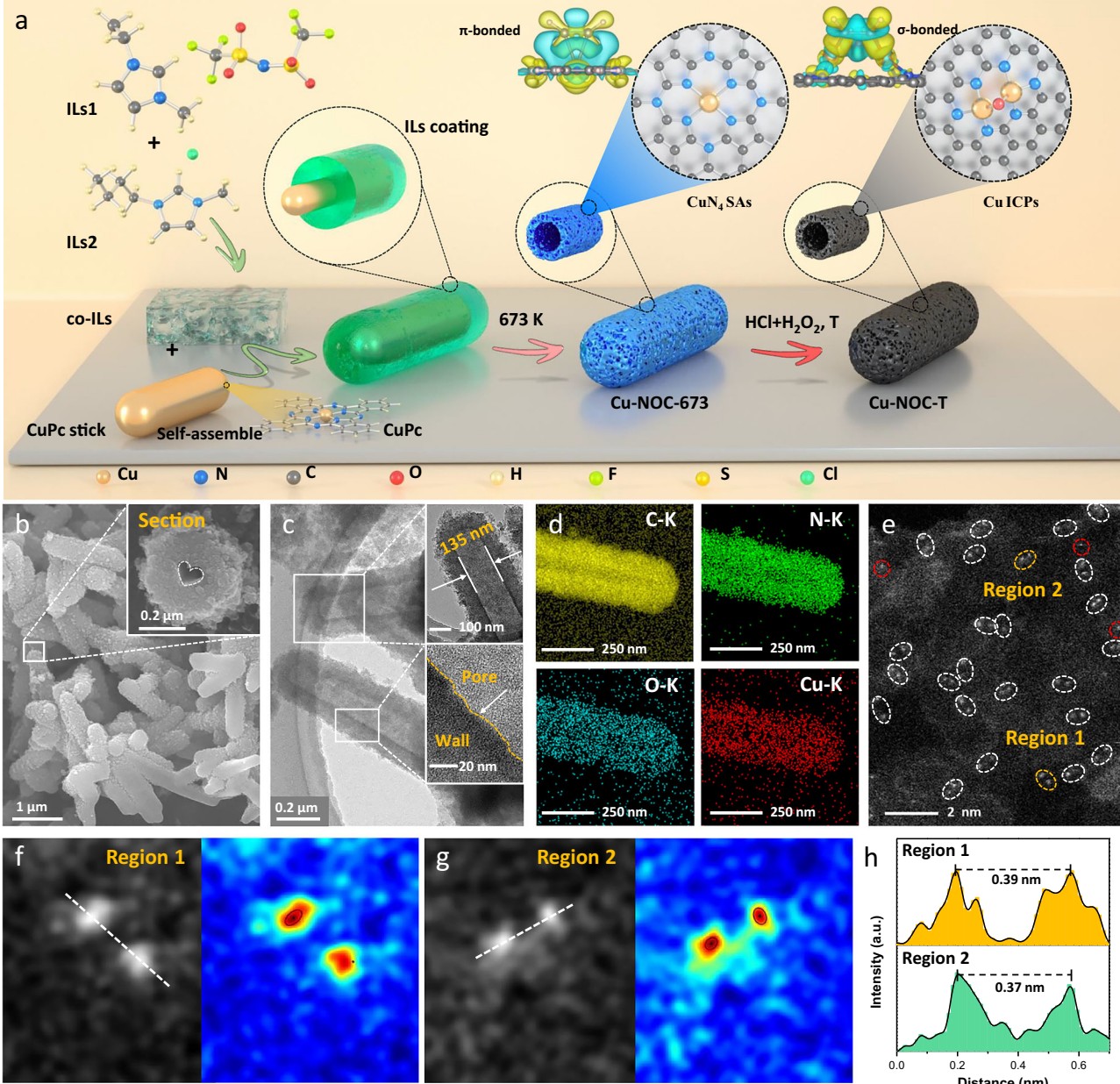

**Fig. 1 | Synthesis procedure and morphological characterization. a** The schematic illustration of the synthetic process for the Cu ICPs catalysts; **b** SEM images of Cu-NOC-973; **c** HR-TEM images of Cu-NOC-973 and **d** corresponding element mapping images showing the distribution of C (yellow), N (green), O (blue), and Cu (red) within the catalyst; **e** HAADF-STEM image of Cu-NOC-973; **f, g** 3D atom-overlapping Gaussian-function fitting maps of Cu ICPs for regions 1 and 2; **h** The corresponding intensity profiles at the regions in (**f, g**).

measured by line-profile analysis (Fig. 1h), the atomic distance between adjacent Cu atoms was measured to be 0.37–0.39 nm with narrow variations for Cu-NOC-973.

Furthermore, atomic-level evolution of Cu structures driven by the Kirkendall effect under different synthesis temperatures was revealed in detail by HAADF-STEM analysis (Supplementary Fig. 13). At a pyrolysis temperature of 673 K, local regions with significant Cu atom aggregation can be observed, although no distinct nanoparticles are formed, the Cu atom density is calculated to be 13.01 Cu/nm². This is primarily because, at 673 K, CuPc does not undergo pyrolysis-induced agglomeration but instead exhibits an accumulation state between CuPc molecules due to insufficient migration driving force. As the temperature of the secondary heat treatment increases, further diffusion and migration of Cu atoms occur, reducing the Cu atom density to 11.10 Cu/nm², at which point numerous Cu atom pairs begin to

appear. When the treatment temperature is raised to 973 K and 1173 K, the Cu atom density decreases further to 7.20 Cu/nm² and 2.63 Cu/nm², respectively, resulting in the formation of distinct Cu ICPs structures on the catalyst surface. In conclusion, the two-step annealing combined with coordination etching successfully synthesized a well-defined hollow nanocapsule-supported Cu ICPs catalyst. Therefore, differences in treatment temperatures result in varying degrees of Kirkendall migration, ultimately determining the dispersion state of Cu. Additionally, the substrate universality of the synthesis method was confirmed through extended experiments using various metal phthalocyanine precursors. It was demonstrated that FePc, CoPc, NiPc, and ZnPc precursors could all form ICPs structures during the synthesis process. As shown in Supplementary Fig. 14, the HAADF-STEM images and the corresponding 3D atomic overlap Gaussian fitting maps indicate the successful construction of uniform integrated metal

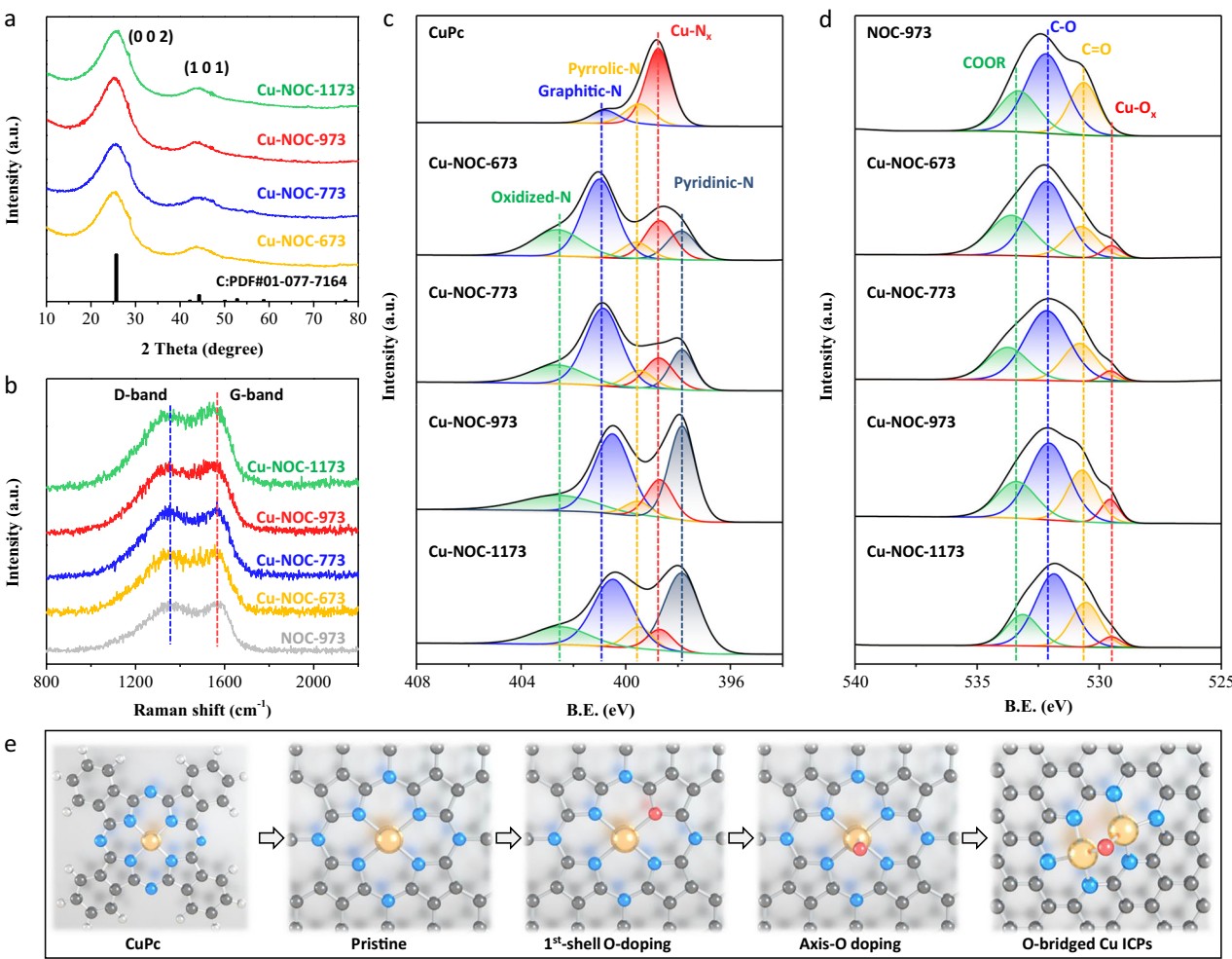

**Fig. 2 | Surface textural characterization.** Microtexture and surface chemical states of both Cu-NOC-973 and NOC-973 with CuPc as reference. **a** XRD patterns; **b** Raman spectra. The experimental and fitted high-resolution XPS spectra of (**c**) N 1*s*, and **d** O 1*s*. **e** The scheme of proposed coordination structure of Cu species.

pairs on the catalyst surface, further demonstrating the generality of this synthesis strategy.

## Coordination and chemical structure of Cu ICPs

The synthesized Cu-NOC series catalysts were systematically characterized using X-ray diffraction (XRD), Raman, and X-ray photoelectron spectroscopy (XPS) to elucidate the evolution of their crystal and defect structures, as well as elemental composition and chemical states. XRD results (Fig. 2a) show broad diffraction peaks at 26° and 43°, corresponding to the 002 and 101 planes of graphite, indicating an amorphous structure[22]. The absence of Cu-related diffraction peaks suggests that Cu is uniformly dispersed, as has been confirmed by HAADF-STEM results in Fig. 1e. Raman spectroscopy results (Fig. 2b and Supplementary Table 2) reveal that all catalyst samples exhibit an $I_G/I_D$ ratio close to 1 at synthesis temperatures below 1173 K, indicating minimal variation in defect concentration. However, at a synthesis temperature of 1173 K, the G-band signal significantly increases, indicating a higher degree of graphitization and a lower defect concentration[31]. Considering the critical role of defects in catalysis, it can be inferred that the catalytic performance of Cu-NOC-1173 would significantly decline compared to Cu-NOC-973.

Figure 2c (Supplementary Fig. 15) shows the N1s XPS results of the Cu-NOC series catalysts, with pure CuPc serving as a reference. The CuPc sample predominantly contains Cu-N$_x$ and a small amount of pyrrolic-N species. After the two-step thermal activation, the Cu-NOC series catalysts exhibit not only pyrrolic-N and Cu-N$_x$ species but also

pyridinic-N, graphitic-N, and oxidized-N species[9,30,34], which are derived from ILs, as confirmed by the NOC-973 characterization results in Supplementary Fig. 16. Among all samples, graphitic-N species are dominant, with the content of oxidized-N species decreasing and pyridinic-N species significantly increasing as the thermal treatment temperature rises. Consequently, the Cu-NOC-973 and Cu-NOC-1173 catalysts demonstrate enhanced thermal stability due to increased graphitization. The O1s XPS results in Fig. 2d show the presence of C-O, C=O, and COOR species generated by the thermal polymerization of the binary ILs, along with noticeable Cu-O$_x$ species[28,35]. Since CuPc molecule typically undergoes structural changes above 923 K, the formation of Cu-O$_x$ species is likely due to H$_2$O$_2$/HCl acid etching and the second thermal activation, which removes excess unreacted and aggregated metals and further modulates the coordination atoms of Cu sites. The highest content of Cu-O$_x$ species is observed in the catalyst treated at 973 K. Based on the N1s and O1s XPS data, it can be inferred that Cu is co-coordinated with N and O in the catalysts. Given the initial structure of CuPc, it can be deduced that after the first thermal activation at 673 K, the catalyst primarily maintains a Cu-N$_4$ coordination structure. Following the second thermal activation, Cu-O coordination occurs and can manifest in two possible forms: first-shell coordination (1st shell O-doping) and axial coordination (axis O-doping), as illustrated in Fig. 2e. Considering that the Cu atoms form a dual-atom structure with an interatomic distance of approximately 0.37 nm—closely matching the Cu-O-Cu distance in CuO (0.373 nm)[36]— it can be preliminarily inferred that the Cu atoms may be coordinated

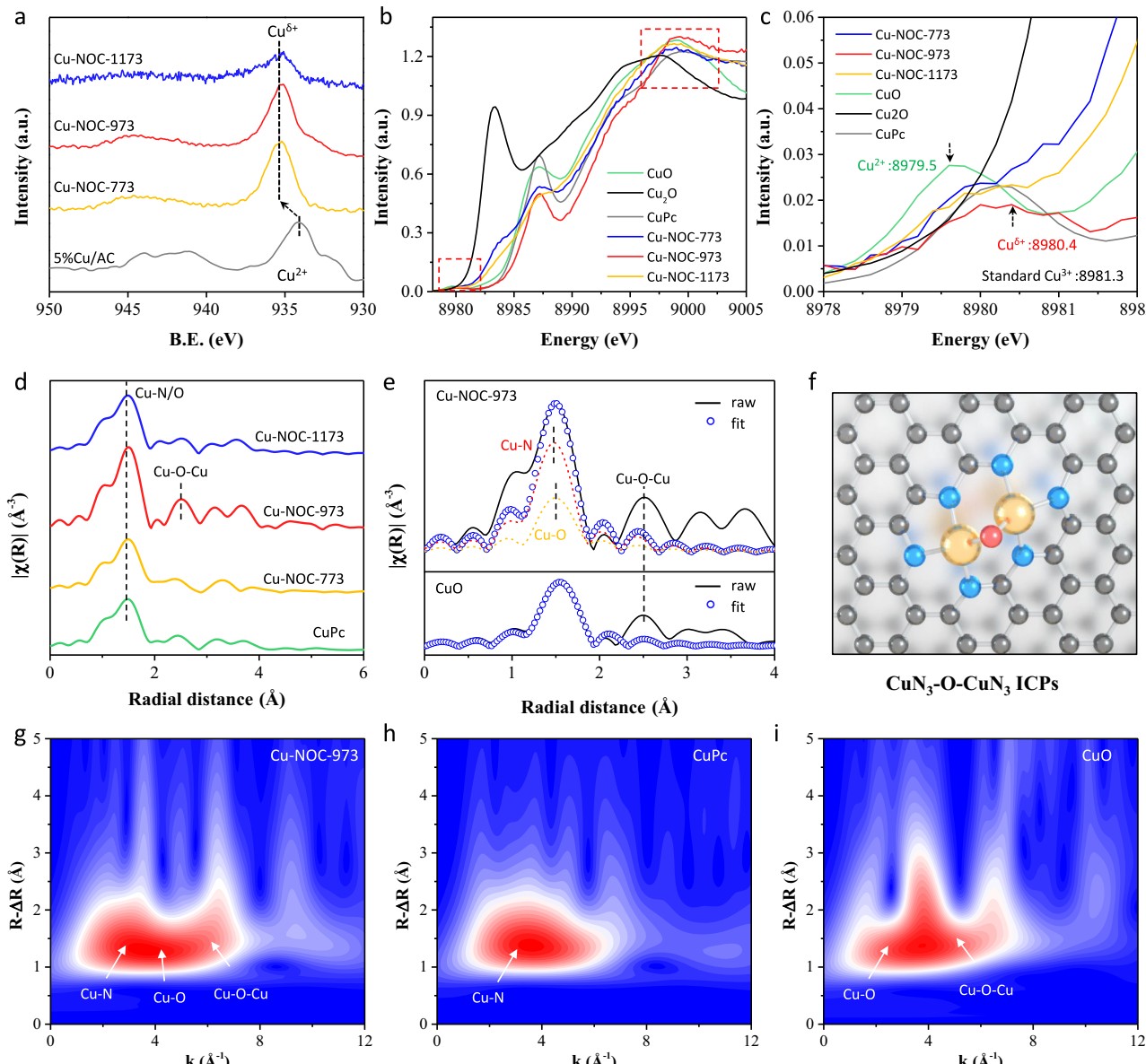

**Fig. 3 | Coordination structural analysis. a** High-resolution XPS spectra of Cu 2p for investigated catalysts; **b** Cu Kβ HERFD-XAS;(**c**) Partial enlarged of the pre-edge region for Cu Kβ HERFD-XAS; **d** Cu K-edge $k^3$-weighted Fourier transform (FT) spectra; **e** R space fitting details of Cu-NOC-973 catalysts with CuO as reference; **f** Proposed Cu coordination structure in Cu-NOC-973; Wavelet transforms (WT) of **g** Cu-NOC-973, **h** CuPc, and **i** CuO.

via an oxygen bridge, and X-ray absorption spectroscopy (XAS) measurements were next conducted to verify this hypothesis.

We further determined the electronic and coordination structures of Cu atoms in Cu-NOC series catalysts by XPS and XAS measurements. Due to the very low content of Cu in the catalysts, multiple scans were performed and then merged for XPS characterization, with more than 10 scanning circles per catalyst. According to Fig. 3a and Supplementary Fig. 17, the XPS spectra of $CuCl_2$/AC catalyst exhibit a dominant peak at binding energies (B.E.) of 933.1 eV ($2p_{3/2}$), which can be assigned to the representative $Cu^{2+}$ peak. For Cu-NOC-973 catalyst, the deconvoluted peaks at 935.4 (Cu $2p_{3/2}$) and 955.2 (Cu $2p_{1/2}$) eV are ascribed to the Cu-$N_x$ species, while the deconvoluted peaks at 944.2 and 963.8 eV are assigned to shake-up satellite peaks (abbreviated as "sat."). It should be noted that these B.E. are much higher (ca. 1.5–2 eV) than that of $Cu^{2+}$ species in $CuCl_2$/AC, indicating the formation of high-valent $Cu^{\delta+}$ ($\delta > 2$) sites[37]. Cu LMM Auger spectra of Cu-NOC-973 (Supplementary Fig. 17g) exhibit characteristic kinetic energy at

around 915.6, 917.5, and 920.3 eV, respectively, suggesting the presence of $Cu^+$, $Cu^{2+}$, and $Cu^{3+}$ species. Furthermore, the Cu 2p XPS and Cu LMM Auger spectra for Cu-NOC-773, and Cu-NOC-1173 also show the presence of $Cu^{3+}$ characteristic peaks, confirming the formation and retention of $Cu^{3+}$ sites during the synthesis process.

To corroborate the oxidation state assignments of the Cu catalysts, and specifically the $Cu^{3+}$ character of Cu-NOC-973 catalysts, Cu Kβ high-energy-resolution fluorescence detection (HERFD) XAS was performed for the series of Cu-NOC catalysts. This detection technique records the Cu K-edge absorption spectrum by monitoring a 3p-1s fluorescence signal with a large, high-resolution emission spectrometer (ca. 1 eV resolution). The resultant narrower absorption linewidth is determined by the longer 3p core−hole lifetime, as opposed to the shorter 1s core−hole lifetime in traditional XAS measurements. As shown in Fig. 3b, the absorption threshold energies of Cu-NOC series samples are positioned between those of selected $Cu^{2+}$ and $Cu^{3+}$ standard samples, that has previously been

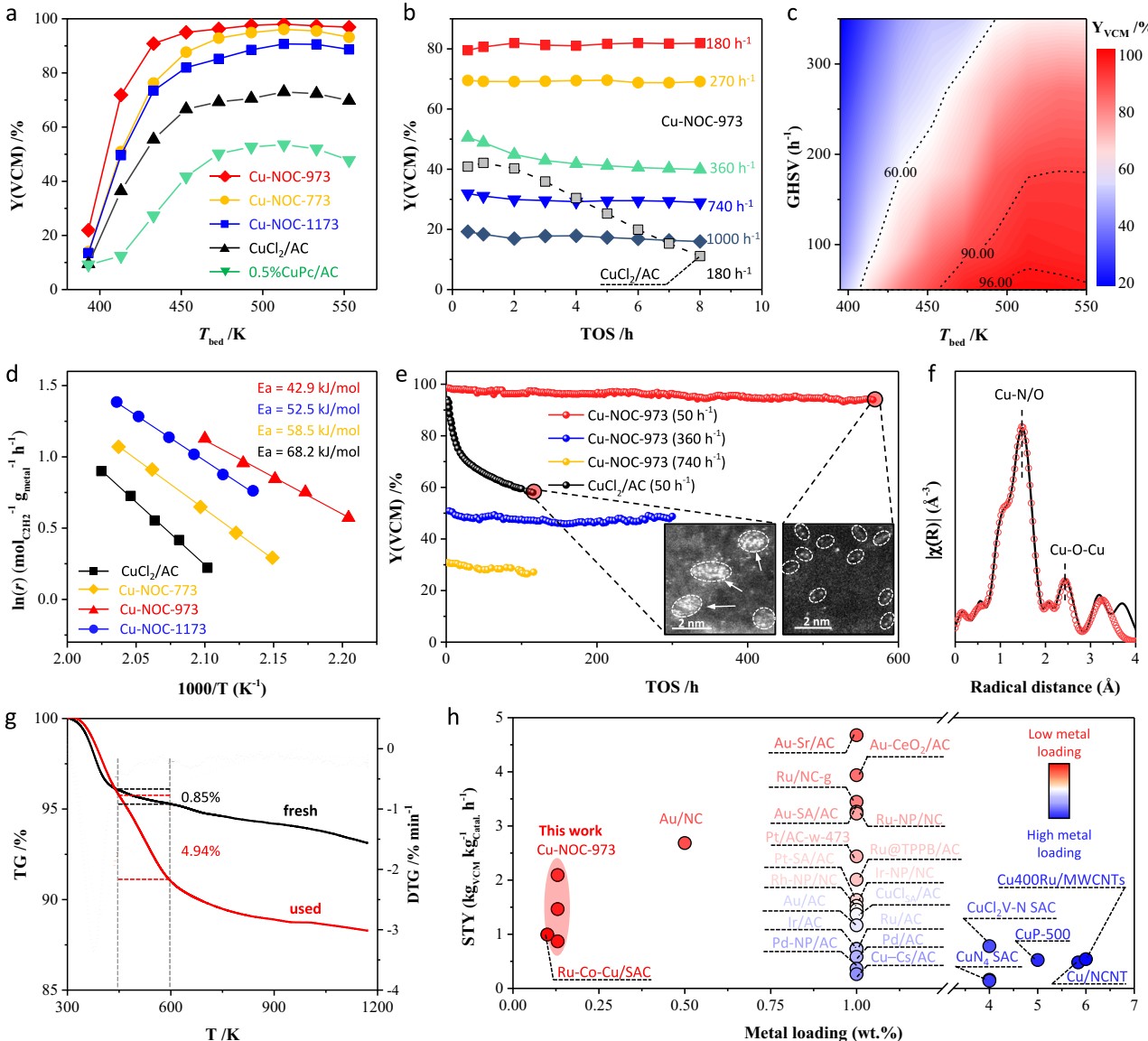

**Fig. 4 | Catalytic performance evaluation.** Catalytic performance of Cu-NOC series catalysts with CuCl₂/AC and CuPC/AC as reference, expressed as the yield of vinyl chloride monomer Y(VCM) at **a** 393–553 K, 50 h⁻¹ and **b** 493 K, 180–1000 h⁻¹; **c** The reaction window for Cu-NOC-973 (393–553 K, 50–360 h⁻¹); **d** Apparent activation energy of the Cu-NOC series catalysts with CuCl₂/AC as reference; **e** Long-term stability of Cu-NOC-973 at 493 K with a GHSV(C₂H₂) of 50 h⁻¹, and corresponding HAADF-STEM images for the used catalysts; **f** XAFS curves fitting in R space and **g** TG and DTG analysis for used Cu-NOC-973 catalysts; **h** Comparison of *STY* of acetylene hydrochlorination catalyzed by Cu-NOC-973 catalysts and by reference metal catalysts (Supplementary Fig. 23). VCM selectivity for investigated catalysts in this work were >99.5% unless otherwise noted.

reported by Liu and Geoghegan et al.[37,38] At 8979.5 eV, a weak pre-edge feature is observed in Cu²⁺ standard samples, corresponding to a dipole-forbidden $1s$-$3d$ transition into the single d electron hole of the $d^9$ Cu²⁺ center. A weak pre-edge feature is observed at 8981.3 eV for Cu³⁺ standard samples, which is approximately 1.8 eV higher than the established Cu²⁺ pre-edge energy of Cu²⁺ standard samples and consistent with previous reports for Cu³⁺ complexes. For Cu-NOC series samples, the pre-edge features are observed between 8979.5 and 8981.3 eV (Fig. 4b and Supplementary Fig. 18a), suggesting that there are abundant Cu³⁺ species in the catalysts. Meanwhile, the observed shifts in the white line energies for Cu-NOC series catalysts between approximately 8994 (Cu²⁺) and 9000 (Cu³⁺) eV are consistent with the increasing copper oxidation state (Supplementary Fig. 18b and Supplementary Table 3), further validating the coexistence of mixed-valence Cu²⁺-Cu³⁺ species.

Moreover, the quantitative least-squares fitting of extended X-ray absorption fine structure (EXAFS) was performed. By comparing the EXAFS fitting curve and fitting parameters of CuPc in Fig. 3d (Supplementary Fig. 19) and Supplementary Table 4, the first shell peak at 1.5 Å is assigned to single Cu atoms coordinated with four N/O atoms as the Cu-N/O coordination in Cu-NOC series catalysts. The coordination numbers (CNs) of Cu-N in Cu-NOC-773, Cu-NOC-973, Cu-NOC-1173 and CuPc are determined to be about 3.57, 3.03, 2.95, and 4.06, respectively. While the CNs of Cu-O in Cu-NOC-773, Cu-NOC-973, and Cu-NOC-1173 are determined to be about 0.54, 0.93, and 0.56 (Supplementary Table 4), respectively. It is noted that the bond length of Cu-N in Cu-NOC-973 (1.908 Å) is significantly shorter than that of CuPc (1.941 Å), which is mainly caused by the enhanced overlap of the electron cloud due to O-bridge coordination. As depicted in Fig. 3e, the experimental results match well with the fitting model, confirming that

the Cu atom of Cu-NOC-973 is coordinated with three N and one O atoms, i.e., $CuN_3O_1$. Furthermore, the wavelet transform (WT) investigation was conducted, which can provide powerful resolution in k and R space and distinguish the backscattering atoms. The WT contour plot of Cu-NOC-973 shows maximum intensities of approximately 3.5, 3.8, and 6.5 $Å^{-1}$, corresponding to Cu-N, Cu-O, and Cu-O-Cu bonds, respectively, as calibrated using CuPc and CuO in Fig. 3g–i and Supplementary Fig. 20. While no maximum intensity indexed to Cu-Cu contribution can be detected compared with the plot of Cu foil. Based on the combined XAS and XPS observations, it can be confirmed that the CuPc furnishes the typical $Cu-N_x$ framework, while high-temperature treatment and coordination-etching induce the formation of Cu-O coordination, resulting in the final $CuN_3-O-CuN_3$ ICPs structure, as illustrated in Fig. 3f. Because of the synergies gained from abundant $Cu-N_x$ active sites and enhanced electronic structure (d/p electrons) delocalization by O-bridging, it is expected that the Cu-NOC-973 catalysts would have significantly improved catalytic activity.

## Catalytic performance evaluation

The intrinsic activities of the ICPs catalysts were further investigated, as shown in Fig. 4. Initially, to find the optimal reaction temperature, a temperature ramp experiment was first carried out. To eliminate the influence of catalyst deactivation, each data point was collected in a separate experiment. Acetylene hydrochlorination reactions across the Cu-NOC catalyst series were evaluated in a fixed-bed quartz reactor (Supplementary Fig. 21) at bed temperatures ($T_{bed}$) ranging from 393 to 553 K and a gas hourly space velocity based on acetylene ($GHSV(C_2H_2)$) of 50 $h^{-1}$. The activity of catalysts progresses the sequence of Cu-NOC-973 > Cu-NOC-773 > Cu-NOC-1173 > CuCl$_2$/AC > 0.5 wt%CuPc/AC. As shown in Fig. 4a, the Cu-NOC series catalysts achieved their highest yields at 493 K, which was subsequently selected for further investigations. In comparison, the CuPc/AC and CuCl$_2$/AC catalysts exhibited significantly lower catalytic activity than the Cu-NOC series. Notably, despite the higher Cu loading in CuPc/AC (0.5 wt %) compared to Cu-NOC-973 (0.1 wt%), its catalytic activity was relatively low (Supplementary Fig. 22), which is mainly attributed to the aggregation of CuPc due to $\pi\cdots\pi$ stacking between molecules along the axial direction. Supplementary Fig. 22a–c shows that CuPc aggregates even with copper loadings as low as 0.1 wt%, which significantly reduces the exposed active sites and, consequently, the catalytic activity. However, this still indicates that CuPc materials have great potential in acetylene hydrochlorination reactions, especially at low Cu loading. Clearly, by converting the CuPc precursor into oxygen-bridged Cu ICPs through a coordination-etching-assisted two-step pyrolysis, the dispersion of Cu sites was significantly enhanced, leading to a substantial improvement in catalytic activity.

We further evaluated the stability of Cu-NOC-973 and 5 wt% Cu/AC catalysts under harsh reaction conditions for acetylene hydrochlorination by conducting reactions at a high bed temperature (493 K) and $GHSV(C_2H_2)$ of 180–1000 $h^{-1}$ for 8 h. As shown in Fig. 4b, the performance of CuCl$_2$/AC decreased significantly at 180 $h^{-1}$, with a 31% loss in activity after 8 h, whereas Cu-NOC-973 exhibited stable performance at 180 $h^{-1}$, and a mild decrease in activity at the range of 360–1000 $h^{-1}$, indicating high catalytic stability of Cu ICPs even in challenging reaction circumstances. Subsequently, we established the catalytic performance window for Cu-NOC-973 at various reaction temperatures and $GHSV(C_2H_2)$ values. As depicted in Fig. 4c, the colors in the two-dimensional maps represent the yield of vinyl chloride monomer (Y(VCM)). The Cu-NOC-973 catalyst achieved nearly 100% Y(VCM) within a temperature range of 493–553 K at low $GHSV(C_2H_2)$ (<100 $h^{-1}$) and maintained over 90% Y(VCM) at temperatures above 453 K with $GHSV(C_2H_2)$ values ranging from 50 to 180 $h^{-1}$. This demonstrates the outstanding catalytic performance of Cu-NOC-973 across a wide temperature and $GHSV(C_2H_2)$ range, with especially superior performance at higher temperatures.

In addition, kinetic experiments determined the activation energy ($E_a$) of the Cu-NOC-973 catalyst to be approximately 43 kJ mol$^{-1}$, which is lower than that of the Cu-NOC-773, Cu-NOC-1173, and CuCl$_2$/AC catalysts (Fig. 4d). This finding further confirms the O-bridge Cu ICPs structure and the resulting high intrinsic activity of the high-valent Cu$^{3+}$ species in catalyzing the hydrochlorination of acetylene. The Cu-NOC-973 catalyst was also tested for long-term stability under industrial reaction conditions (50–740 $h^{-1}$), as shown in Fig. 4e. The Cu-NOC-973 catalyst exhibited a gradual deactivation, maintaining over 94% of its initial activity up to 580 h at 50 $h^{-1}$. Notably, the catalyst maintains excellent long-term stability even at high $GHSV(C_2H_2)$ of 360 and 740 $h^{-1}$, demonstrating its robust catalytic stability under demanding reaction conditions. In contrast, the 5 wt% CuCl$_2$/AC catalyst showed a rapid deactivation rate at 50$^{-1}$, with activity dropping below 60% after approximately 120 h. Note that the XRD, XAS, XPS, HAADF-STEM, BET, and elemental analyses for the used Cu-NOC-973 catalysts are conducted to clarify the local structure of the catalysts after long-term reduction as illustrated in Fig. 4f, g and Supplementary Fig. 24, which clearly indicates that no morphological, structural, or compositional changes take place for Cu sites during reaction. Specifically, XRD and HAADF-STEM investigations verified the presence of only Cu ICPs in the Cu-NOC-973 catalyst employed, whereas many nanoparticles were observed on CuCl$_2$/AC (Supplementary Fig. 25). Meanwhile, there is no discernible difference between the XAS and XPS plots of the used catalysts (Fig. 4f and Supplementary Fig. 24d, e) and those of the fresh catalysts, indicating that the chemical valence of Cu species was maintained during the reaction. As a result, we conclude that the deactivation of Cu-NOC-973 is not caused by changes in the active sites. Instead, there are differences in the porous properties between the fresh and used catalysts (Supplementary Fig. 24f), with the used Cu-NOC-973 showing a significant decrease in surface area and pore volume (502– 363 $m^2 g^{-1}$ and 0.21– 0.17 $cm^3 g^{-1}$, respectively), accompanied by an increase in coke deposition (from 0.85 to 4.94 wt%, Fig. 4g). These findings suggest that coking might lead to the pore blockage, which probably caused the catalyst deactivation for Cu-NOC-973.

It is important to plot the sustainable productivity, in term of space-time yield (STY) based on catalyst packing, to make a fair comparison between different catalysts and understand the effect of acetylene hydrochlorination independent of metal loading. Figure 4h presents the STY of acetylene hydrochlorination for different catalysts relative to their metal loadings, enabling a comparison between the catalytic activities of Cu ICPs catalysts and state-of-the-art noble metal SACs such as Au, Pd, Pt, Ru, as well as representative copper catalysts[21,22]. The reported STY values for acetylene hydrochlorination over Cu-based catalysts were typically less than 0.6 kg$_{VCM}$ kg$_{Catal.}^{-1}$ h$^{-1}$ at metal loadings above 4 wt%, whereas state-of-the-art noble metal catalysts exhibited STY values exceeding 3 kg$_{VCM}$ kg$_{Catal.}^{-1}$ h$^{-1}$ at a metal loading of 1 wt%[28]. Our catalysts demonstrated significantly higher STY values for acetylene hydrochlorination at low metal loadings (<0.2 wt %) compared to reported Cu catalysts and were comparable to most noble metal catalysts[39]. As shown in Fig. 4h and Supplementary Table 5, traditional Cu-based catalysts are considerably less active than state-of-the-art noble metal SACs, such as Au, Pt, Pd, and Ru, which has historically hindered the replacement of noble metal catalysts with non-noble metal alternatives for acetylene hydrochlorination. Encouragingly, the highly active Cu-NOC-973 catalyst has bridged this significant performance gap between non-noble and noble metal catalysts, breaking the universal standard electrode potential linear scaling relations[19]. The TOF values for Cu-NOC-973 increased from 611 to 869 $h^{-1}$ as the metal loading decreased from 0.1 to 0.01 wt%. These results demonstrate that Cu ICPs with mixed-valence Cu$^{2+}$-Cu$^{3+}$ species constitute a highly active phase for carbon-supported copper catalysts, providing a promising direction for the rational design of high-performance copper hydrochlorination catalysts.

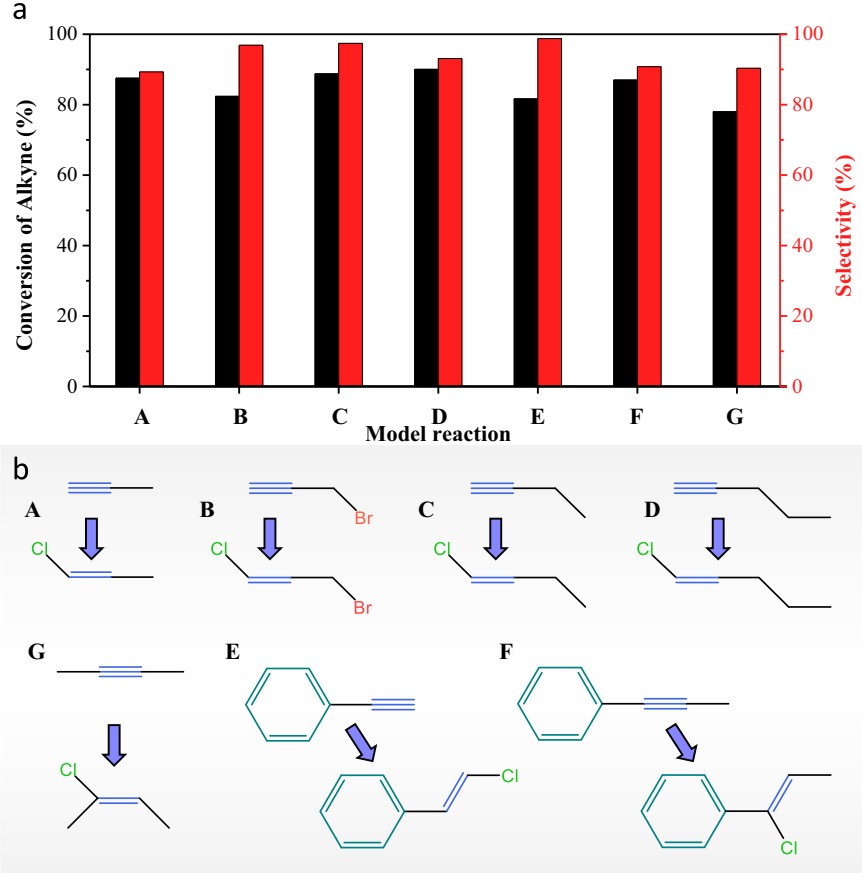

**Fig. 5 | Alkyne selective hydrochlorination. a** Comparative performance of Cu-NOC-973 catalyst in alkynes hydrochlorination and **b** corresponding schematic structures.

Most of the relevant alkynes for the pharmaceutical and fine chemical industries are highly branched compounds with internal triple bonds and additional functionalities. To explore the scope of Cu ICPs systems, the performance of the Cu-NOC-973 catalysts was evaluated in the catalytic hydrochlorination of five terminal alkynes (propyne, propargyl bromide, 1-butyne, 1-pentyne and phenylacetylene) and two internal alkynes (2-butyne, phenylpropyne), as shown in Fig. 5 and Supplementary Table 6, several relevant reactions in the production of functional materials and synthesis of intermediate for medicine[18]. The results indicate that the Cu-NOC-973 catalyst exhibits high conversion rates (>75%) for seven compounds containing alkynyl, with selectivity for the reaction products exceeding 90%. Furthermore, the large-scale reactions and catalyst recyclability studies indicated that the Cu-NOC-973 catalyst furnished high stability, with the conversion and selectivity exceeding 90% after five successive cycles (Supplementary Fig. 26). Therefore, the construction of the synergistic integrated catalytic pairs system demonstrates considerable versatility in enhancing the conversion efficiency of alkynes and the selective transformation of target products.

**Selective orbital coupling and reaction mechanism**
Initially, different descriptors are combined with the corresponding Y(VCM) (Supplementary Figs. 27–29). It is clear that the specific surface area, defects and N/O contents of Cu-NOC materials are not the main factor affecting the Y(VCM) (Supplementary Fig. 27). Furthermore, the ratio of oxidized-N, graphitic-N, pyrrolic-N, Cu-N$_x$, pyridinic-N as well as COOR and C=O were not displaying a linear relationship with Y(VCM), while a higher Y(VCM) can be obtained with the presence of more CuO$_x$ species (Supplementary Fig. 29d), which may be attributed to the enhanced electronic structure delocalization by O-bridging. The relationship between the valence state of copper sites

and corresponding catalytic activity was then investigated. As shown in Supplementary Fig. 30, the catalytic activity has a positive dependence on the copper valence, a higher Cu$^{3+}$ fraction led to higher activity, which is likely due to the electron deficient of the Cu $d$-orbital, indicating their potential role in acetylene hydrochlorination. Although Cu$^+$ and Cu$^{2+}$ are constantly mentioned as the active center in literature, the catalytic ability of high-valence Cu$^{3+}$ sites has not been assessed yet. Here, we have shown that Cu ICPs with Cu$^{3+}$ species exhibit outstanding catalytic performance for acetylene hydrochlorination and could be a potential new generation of hydrochlorination catalysts. Further studies are needed to elucidate the origin of the excellent catalytic activity of Cu ICPs and to gain a deeper understanding of its catalytic mechanism.

Furthermore, density functional theory (DFT) calculations were conducted to understand how the O-bridged Cu ICPs species promote the hydrochlorination of acetylene based on the aforementioned atomic structural analyses. To simplify the presentation, six model structures are considered here: (i) an O-bridged Cu ICPs structure (Supplementary Fig. 31 and Supplementary Table 7), representing the high-valent Cu-NOC-973 catalyst; (ii) CuCl and CuCl$_2$ on a graphite surface, hydroxylated at the step edge to stabilize the slabs, representing monovalent and divalent copper catalysts, respectively[14]; (iii) Cu-N$_4$ on a graphite surface, representing typical N-anchored copper catalysts[40]; (iv) CuN$_4$-CuN$_4$ and CuN$_3$O-CuN$_3$O dual Cu atom sites. Adsorption energy analyses show that the adsorption energy of HCl and C$_2$H$_2$ over Cu ICPs is much higher than that over CuCl, CuCl$_2$, Cu-N$_4$, CuN$_4$-CuN$_4$ and CuN$_3$O-CuN$_3$O structures (Fig. 6a), which is consistent with the temperature-programmed desorption (TPD) results (Fig. 6b). The C$_2$H$_2$-/HCl-TPD profile of Cu-NOC-973 shows a higher desorption peak (>650 K) for the dissociative adsorption of C$_2$H$_2$/HCl than that of CuCl$_2$/AC, confirming that the Cu ICPs are more effective

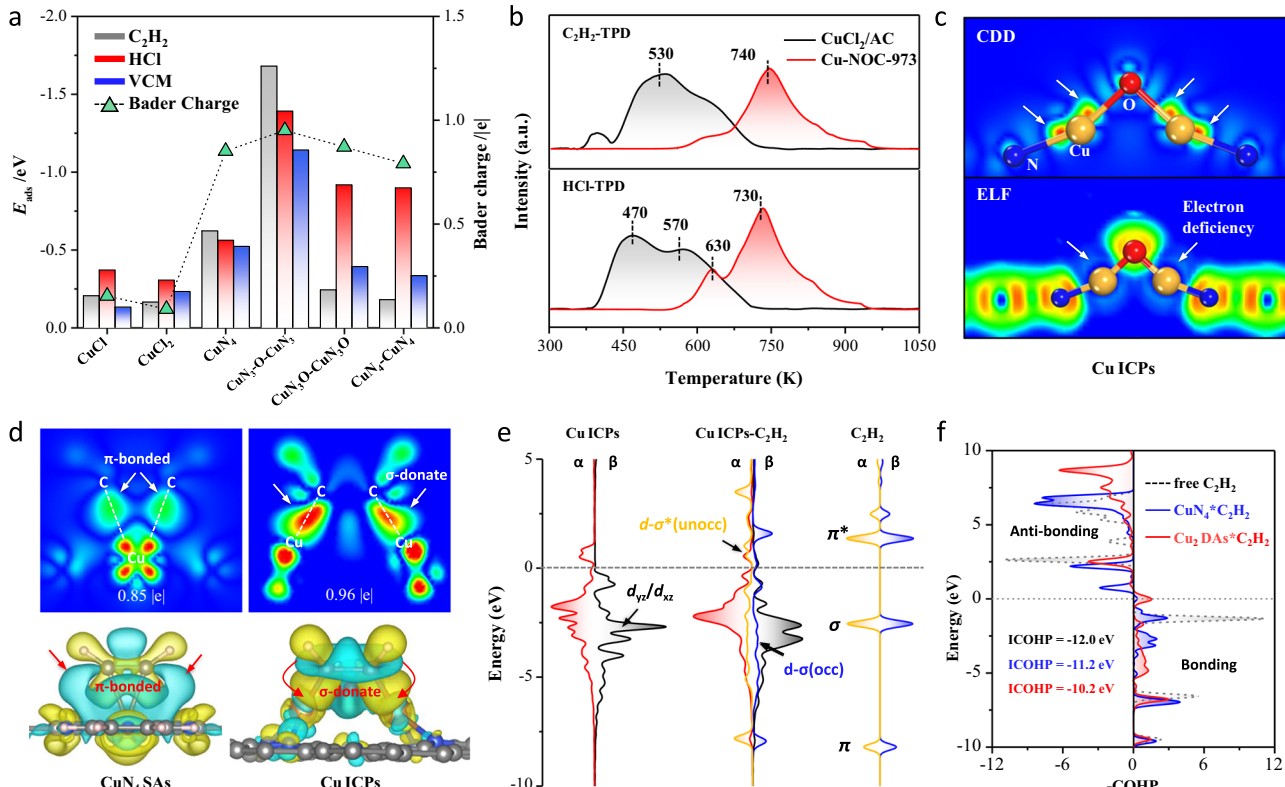

**Fig. 6 | Insight into the catalytic mechanism. a** Adsorption energies of $C_2H_2$, HCl and VCM versus the Cu Bader charge on CuCl, $CuCl_2$, $CuN_4$, $CuN_4$-$CuN_4$, $CuN_3O$-$CuN_3O$ and Cu ICPs species; **b** $C_2H_2$-/HCl-TPD profile over $CuCl_2$/AC and Cu-NOC-973; **c** Charge density difference (CDD) and electron localization function (ELF) on Cu ICPs; **d** Charge density differences of $C_2H_2$ adsorption on Cu ICPs and $CuN_4$ sites; This figure was generated using VESTA program[52], which is

acknowledged; **e** Projected electronic densities of states (PDOS) and schematic illustrations of 3d orbitals of Cu ICPs on, 2p-orbitals of the $C_2H_2$ gas molecule, and their interaction within $Cu_2$-$C_2H_2$ configuration; **f** Crystal orbital Hamilton population (COHP) and integrated COHP (ICOHP) for C-C atomic pairs of $C_2H_2$ adsorption on Cu ICPs, with $CuCl_2$ and $CuN_4$ sites as references.

for substrate adsorption and activation, potentially leading to higher catalytic activity. The electronic structure of the Cu ICPs moiety was also performed and compared with those of CuCl, $CuCl_2$, Cu-$N_4$, $CuN_4$-$CuN_4$, and $CuN_3O$-$CuN_3O$. Bader charge analysis revealed that the Cu site in the Cu ICPs carries a more positive charge (0.96 |e|) compared to other Cu sites, suggesting that the coordinated N and bridge-O atoms attract more electrons from the Cu center, resulting in a higher valence state for Cu in the ICPs. This is further supported by charge density difference (CDD) and electron localization function (ELF) analyses. As illustrated in Fig. 6c and Supplementary Fig. 32, the CDD shows that electrons are transferred from Cu atoms to the coordinated N and O atoms, while the ELF profile indicates a reduced distribution of localized electrons on Cu sites, leading to significantly electron-deficient states of Cu, thereby facilitating better activation of the alkyne substrate. In addition. Compare with the $CuN_4$ site (the planar $D_{4h}$ symmetry), the non-planar $CuN_3$-O-$CuN_3$ configuration displays split $d_{x2-y2}$ peaks and strong $d_{z2}$−O2p hybridization (Supplementary Fig. 33), which leads to electron delocalization and stabilization of the active site. The delocalization arises from the Cu-O-Cu bridging structure, where the oxygen atom mediates electronic communication between the two Cu centers, promoting d-orbital interactions and electronic flexibility.

Since the adsorption energies of reactants are critically important descriptors for catalytic performance. Further calculations were conducted to determine the adsorption configurations of acetylene on $CuN_4$ single-atom and Cu ICPs, and the electron transfer in these configurations was analyzed (as shown in Fig. 6d and Supplementary Fig. 34). Acetylene adopts a $\pi$-adsorption configuration on the classic $CuN_4$ single-atom site, whereas it assumes a di-$\sigma$ adsorption

configuration on the Cu ICPs. Typically, both metal nanoparticles (NPs) and SACs exhibit catalytic activity for alkyne addition, with the di-$\sigma$ and $\pi$ adsorption configurations being more favorable for NPs and SACs, respectively[41]. Because the di-$\sigma$ configuration more readily facilitates addition reactions, NPs generally display higher catalytic activity than SACs. In this case, the synergistic effect between the Cu atoms in the Cu ICPs results in performance akin to that of NPs, enabling di-$\sigma$ adsorption and activation of acetylene molecules. Propelled by these compositional and structural features, it is expected that the ICPs would exhibit distinctly different catalytic properties compared to the SACs in alkynes addition. This explains the origin of the higher catalytic activity observed in the Cu-NOC-973 catalyst. Further analysis (Fig. 6e and Supplementary Table 8) examined the interaction between the projected density of state profiles of the Cu d-orbital in the Cu ICPs and the frontier molecular orbitals of acetylene, revealing strong coupling and orbital hybridization between the acetylene $\sigma$ orbital and the $d_{xz}$/$d_{yz}$ orbitals of the Cu ICPs. Additionally, crystal orbital Hamilton population (COHP) calculations (Fig. 6f) indicated that the integrated COHP (ICOHP) value for the C−C bond in acetylene, after $\pi$ adsorption activation, decreased from −12.0 eV (in the original acetylene) to −11.2 eV, while the ICOHP value after di-$\sigma$ adsorption activation further dropped to −10.2 eV. Generally, more negative ICOHP values correspond to stronger bonding. Apparently, the di-$\sigma$ adsorption configuration is more effective at activating acetylene.

Subsequently, the reaction pathway for acetylene hydrochlorination on the Cu ICPs site was calculated (Fig. 7 and Supplementary Fig. 35), and the reaction energy barriers were compared with those of $CuN_4$, CuCl, and $CuCl_2$. Based on the adsorption energy data, all reactions on SACs ($CuN_4$, CuCl, and $CuCl_2$) follow the E-R

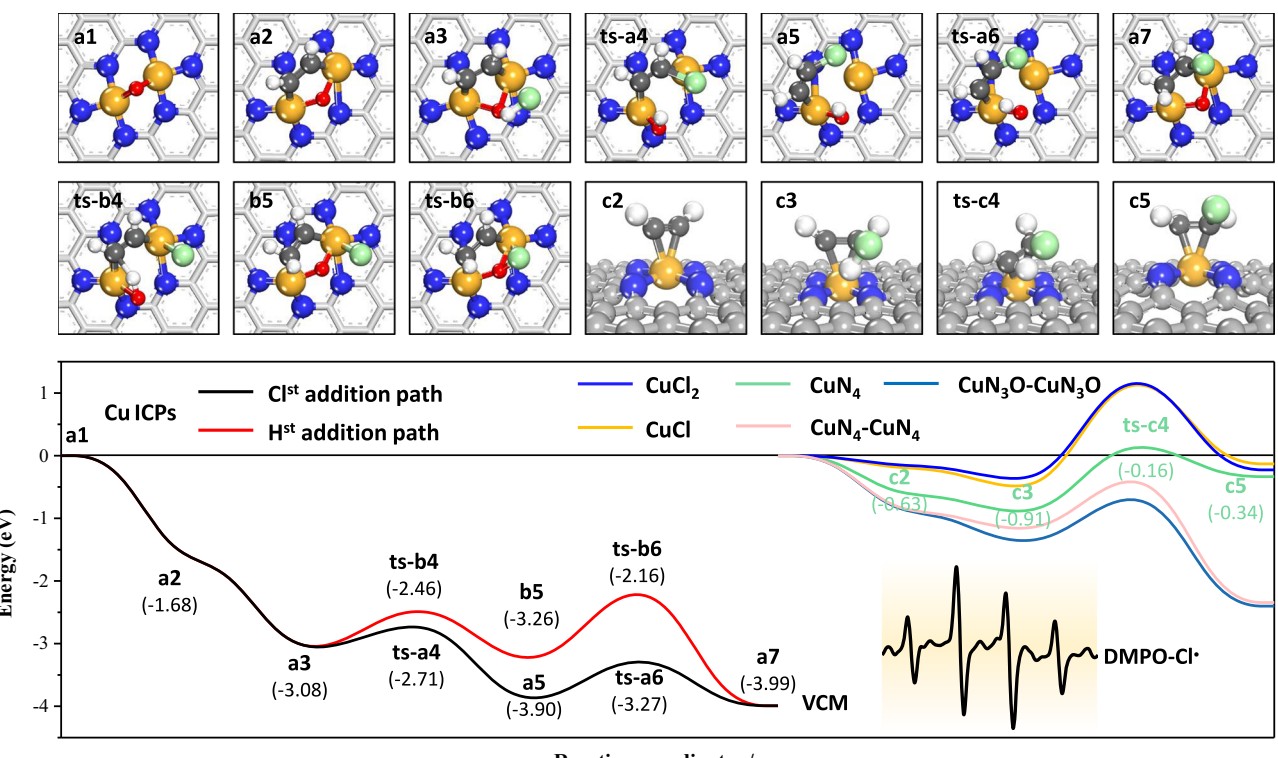

**Fig. 7 | Reaction profiles.** The complete reaction cycle of acetylene hydrochlorination over the Cu ICPs (black and red lines), CuCl (yellow line), CuCl$_2$ (blue line), CuN$_4$ (c2–c5, green line), CuN$_4$-CuN$_4$ (pink line), and CuN$_3$O-CuN$_3$O (cyan line); inset: ESR spectra of DMPO capturing Cl• during HCl adsorption on Cu-NOC-973 catalyst.

mechanism, whereas those on dual-site ICPs adhere to the L-H mechanism, owing to the integrative active sites. In this scenario, C$_2$H$_2$ adsorbs on the Cu atom pairs within the Cu ICPs, while HCl primarily adsorbs on the oxygen atoms. At the initial adsorption stage (Fig. 7, a2), the adsorption energy of acetylene on Cu ICPs (−1.68 eV) is significantly higher than that on CuN$_4$ (−0.62 eV), CuCl (−0.20 eV), and CuCl$_2$ (−0.16 eV), indicating the significant synergistic effect between the integrated Cu atoms. Similarly, the adsorption energy of HCl on Cu ICPs (−1.339 eV) is also significantly higher than that on CuN$_4$ (−0.56 eV), CuCl (−0.37 eV), and CuCl$_2$ (−0.30 eV). Notably, HCl dissociates to release Cl radicals upon adsorption, which has been demonstrated to furnish higher activity[24,42]. The formation of Cl radicals[43] can be confirmed by the electron paramagnetic resonance data shown in the inset of Fig. 7 and Supplementary Fig. 36. Due to the dissociation of HCl, the addition reaction may proceed through various intermediates, categorized into two pathways based on the sequence of addition: Cl-first addition (Cl$^{st}$ addition path, black line in Fig. 7) and H-first addition (H$^{st}$ addition path, red line), evidently, the Cl$^{st}$ addition path has a lower reaction potential energy surface. More importantly, the reaction energy barrier for the rate-determining step of the Cl$^{st}$ addition path on Cu ICPs is only 0.63 eV (Supplementary Table 9), significantly lower than that on CuN$_4$ (0.92 eV), CuCl (1.7 eV), and CuCl$_2$ (1.6 eV), confirming a significant synergistic effect between the integrated Cu atom pairs and bridging oxygen atom. Additionally, the adsorption energy of VCM on Cu ICPs (−1.14 eV) is substantially higher than on CuN$_4$ (−0.52 eV), CuCl (−0.13 eV), and CuCl$_2$ (−0.23 eV). This strong adsorption may inhibit the release of toxic VCM intermediates during the reaction, ultimately leading to carbon deposition on the catalyst surface.

Acetylene hydrochlorination reaction with an ambient pressure on Cu-NOC-973 at various temperatures was further studied by time-resolved in-situ diffuse reflectance Fourier-transform spectroscopy (DRIFTS). As a comparison, the in-situ DRIFTS of acetylene hydrochlorination was first conducted on CuCl$_2$/AC catalysts. As

shown in Fig. 8a, the vibrational features of gas phase C$_2$H$_2$ with fine structures appear at 3302, 3249 cm$^{-1}$ [9,44]. Meanwhile, the vibrational features of gas-phase VCM are also visible at 3078 cm$^{-1}$ in the spectra and increases with the reaction temperature, evidencing the occurrence of acetylene hydrochlorination reaction. Additionally, characteristic peaks for π-bonded and di-σ-bonded HC≡CH appeared at 2150 and 1590 cm$^{-1}$ [9,44,45], respectively, although the signal intensity for both was quite weak. Furthermore, we conducted the in-situ DRIFTS characterization of Cu-NOC-973 at various operation temperatures. As shown in Fig. 8b, the intense characteristic vibration of di-σ-bonded HC=CH species appeared at 1600 cm$^{-1}$, significantly higher than CuCl$_2$/AC catalysts. The strong di-σ-bonded HC=CH indicates the partial dissociative chemisorption of acetylene, which aligns with the DFT calculation results. It is noting that the adsorption intensity of VCM species (3050/3084 cm$^{-1}$) gradually increases with the rise of temperature, meaning the higher acetylene conversion to VCM. And the characteristic peak intensity of acetylene at 3318 and 3260 cm$^{-1}$ was at the middle value, which is conducive to the progress of the reaction. The above in-situ DRIFTS results clearly identify partial dissociative chemisorption of di-σ-bonded HC=CH on Cu-NOC-973 catalysts during the acetylene hydrochlorination reaction, which is responsible for the high activity of the catalysts and is a key intermediate species for the preparation of copper catalysts with high activity.

## Discussion
In summary, this study has demonstrated a new high-valent Cu ICPs catalyst, which has comparable catalytic activity for acetylene hydrochlorination with noble-metal-based catalysts. Benefiting from a rational, optimized protocol via structure functionalities and electronic control of Cu ICPs, the unique O-bridging Cu ICPs empowers the Cu-NOC with enhanced intrinsic activity and excellent catalytic stability at low metal loading. Moreover, a combination of experimental and theoretical techniques revealed that the significant role of the high-valent Cu atom pair and the bridging-O species in controlling the

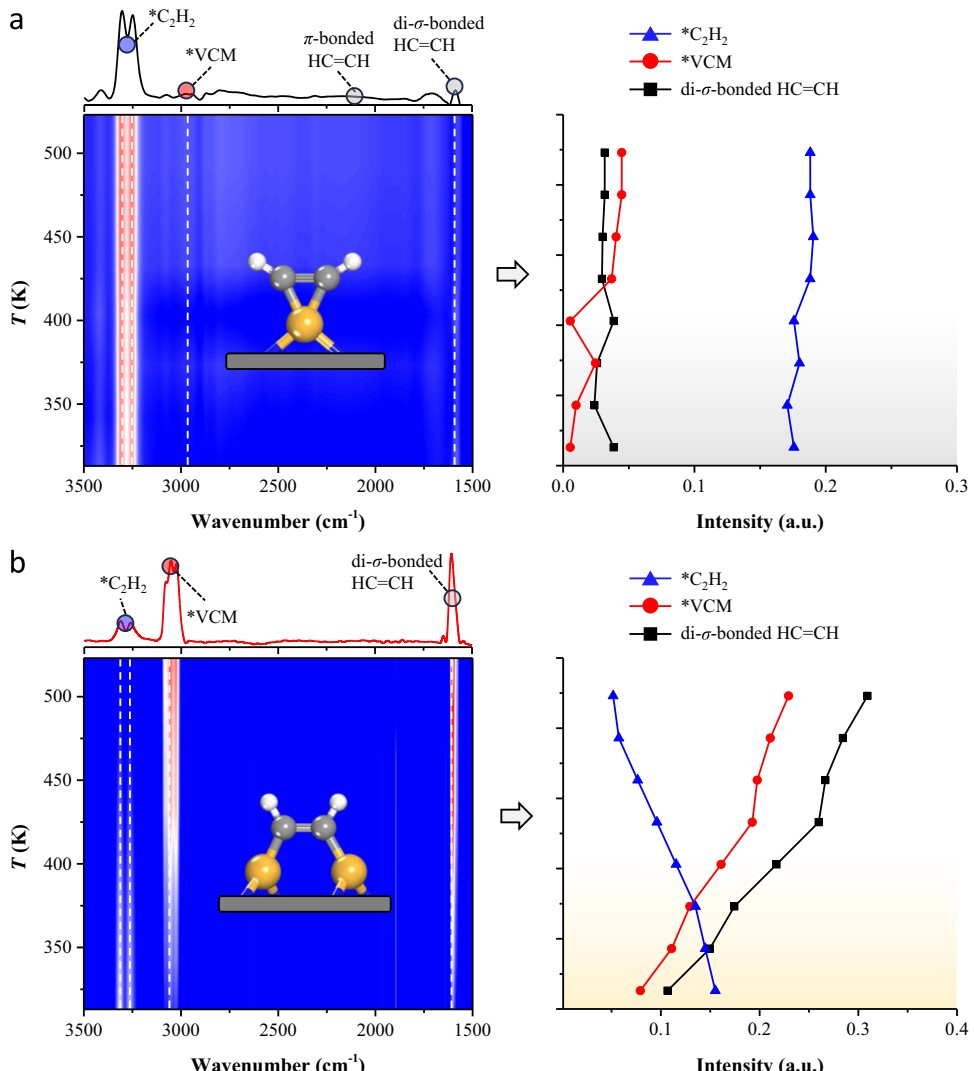

**Fig. 8 | Experimental mechanistic evidence.** In-situ DRIFTS spectra of the (**a**) CuCl$_2$/AC and **b** Cu-NOC-973 catalysts during the heating ramp conditions. Conditions: $W_{cat}$ = 0.015 g, heating rate 5 K min$^{-1}$, $T$ = 273–523 K, GHSV(C$_2$H$_2$) = 90 h$^{-1}$, HCl:C$_2$H$_2$ = 1.05:1.

adsorption pose of C$_2$H$_2$ and strengthening the selective orbital coupling between them. These insights provide valuable guidance for modifying or designing better Cu-based catalysts via multiple-sites synergetic and valence approaches. This work provides a successful paradigm to strengthen intrinsic catalytic activity for the selective addition of alkynes via structure functionalities and electronic control of active sites, which can be extended to design and optimize other catalysts with high efficiency.

## Methods
### Catalyst preparation
The O-bridged Cu ICPs catalyst (Cu-NOC-973) was synthesized using a template casting method, involving a two-step thermal treatment and coordination etching. The process primarily includes the self-assembly of CuPc, coating and polymerization of a binary mixed ILs, followed by coordination etching and carbonization. Specifically, (CuPc, 0.54 g, Aladdin, 99%) was first dispersed in an anhydrous ethanol solution and stirred at 323 K for 2 h to self-assemble into uniformly sized CuPc nanorods. Next, 1-butyl-3-methylimidazolium chloride ([BMIm]Cl, 10 g, Aladdin, 99.5%) was added to 1-ethyl-3-methylimidazolium bis[(-trifluoromethyl)sulfonyl]imide ([EMIm]NTf$_2$, 50 g, Aladdin, 99.5%) and stirred at 323 K for 30 min, followed by the addition of the prepared CuPc solution. The mixture was then sonicated for 20 min and

vigorously stirred (300 rpm) at 273 K for 24 h to complete the coating process. The mixed sample was then placed in a tube furnace and heated at 773 K for 2 h under a nitrogen atmosphere to complete the polymerization of the ILs. Subsequently, the polymerized sample was treated with a mixed solution of H$_2$O$_2$ and hydrochloric acid (1:3) to remove agglomerated Cu particles and perform coordination etching. The treatment process involved boiling the mixed solution, adding the polymerized sample, and treating it for 3 h, followed by filtration and washing with deionized water until neutral. Finally, several temperatures (673, 773, 973, and 1173 K) were selected to determine the optimal carbonization temperature for preparing Cu-NOC, as the oxygen atom doping concentration depends on the carbonization temperature. These materials are denoted as Cu-NOC-773, Cu-NOC-973, and Cu-NOC-1173. Before measuring catalytic activity, the obtained Cu-NOC was ground and sieved (0.6–0.8 mm). For comparison, a copper-free non-metal material, NOC-973, was synthesized using the same process as Cu-NOC-973, except CuPc was not used as a template. To eliminate metal contaminants, commercial activated carbon (AC, Norit ROX 0.8) was immersed in aqua regia at room temperature. After filtration and washing with deionized water, the activated carbon was dried in ambient air at 393 K for 12 h and then used to prepare CuCl$_2$/AC (5 wt%) and CuPc/AC (Cu: 0.5 wt%) catalysts by an impregnation method using CuCl$_2$ and CuPc as precursors, respectively.

## Catalyst evaluation

In a continuous-flow fixed-bed micro-reactor at atmospheric pressure, steady-state hydrochlorination of acetylene was investigated. Digital mass-flow controllers (Horiba) were used to feed the reactant gas streams of $C_2H_2$, HCl, and $N_2$ into the mixing unit, which had a pressure indicator. The catalyst was loaded into a quartz micro-reactor with a 15 mm inner diameter, and it was then placed in an oven. The reaction temperature was managed using a K-type thermocouple mounted in a coaxial quartz thermowell, with the tip placed in the middle of the catalyst bed. Before testing, the catalyst was brought up to temperature ($T_{bed}$ = 393–673 K) in flowing $N_2$ and maintained there for at least 10 minutes. The reaction mixture was then fed at a $C_2H_2$ gas hourly space velocity (GHSV($C_2H_2$)) of 50-740 $h^{-1}$. Acetylene hydrochlorination reaction kinetics over Cu-NOC and Cu/AC catalysts were investigated at <20% conversion in the temperature range of 473–573 K with a total flow rate of 22 $cm^3 min^{-1}$. The reaction orders of reactants ($n_{C2H2}$, $n_{HCl}$) were explored at reactant concentrations of 10–30 vol% balanced with $N_2$. Carbon-containing compounds ($C_2H_2$, VCM, and byproducts) were detected online using a gas chromatograph (FULI INSTRUMENTS F60 with an FID detector). The yield of vinyl chloride (VCM), $Y_{VCM}$, was calculated using Eq. 1,

$$Y_{VCM} = \frac{n_{VCM}^{outlet}}{n_{C_2H_2}^{inlet}} \times 100\% \qquad (1)$$

where $n_{C_2H_2}^{inlet}$ and $n_{VCM}^{outlet}$ are the molar flows of acetylene and VCM at the inlet and outlet of the reactor, respectively. VCM selectivity for investigated catalysts in this work were >99.5% unless otherwise noted.

The space time yield (STY) was determined according to Eq. 2,

$$STY, kg_{VCM} h^{-1} kg_{Catal.}^{-1} = \frac{VCM\ generated}{W_{Catal.} * t_{React.}} \qquad (2)$$

where $W_{catal.}$ denotes the catalyst mass and $t_{React.}$ denotes the reaction time.

For the gaseous reactants (propyne, 1-butyne, and 2-butyne), the reactions were performed in a fixed-bed reactor. The reactant gas stream (30 $h^{-1}$), diluted to 1% concentration in nitrogen, was thoroughly mixed with hydrogen chloride (HCl) in a mixing chamber before being introduced into the fixed-bed reactor containing the Cu-NOC-973 catalyst. The effluent gases were analyzed using an online gas chromatography-mass spectrometry (GC-MS) system. For liquid-phase reactants, the reactions were conducted in a high-pressure autoclave equipped with a polytetrafluoroethylene (PTFE) liner. After transferring 0.1 mol of reactant into the reactor vessel with Cu-NOC-973 catalysts ($n_{Cu}$:$n_{reactant}$ = 1:10), HCl gas was introduced to achieve a system pressure of 0.3 MPa. The reaction temperature was then elevated to the predetermined target temperature to initiate the reaction. The products were analyzed by high-performance liquid chromatography coupled with mass spectrometry.

## Catalyst characterization

Using Cu K radiation (=0.1541 nm) and a PANalytical Empyrean powder diffractometer, XRD patterns were captured. Working voltage and current were 40 kV and 40 mA, respectively. The patterns were gathered with a step size of 0.0262 and a 2 range from 10 to 80°.

Using an FEI Titan 80-300 electron microscope outfitted with a high-brightness X-FEG electron source and a CEOS aberration corrector for the probe-forming lenses, HR-TEM was carried out.

The morphologies of samples were investigated by the field emission scanning electron microscopy (FE-SEM) images obtained by a NOVA NanoSEM 450 instrument operated at the beam energy of 5 kV.

A Kratos AXIS Ultra DLD instrument with a hemispherical capacitor electron-energy analyzer was used to capture X-ray

photoelectron spectra (XPS) using monochromatic Al-Kα radiation produced by an electron beam running at 15 kV.

Measurements of the Cu-K X-ray edge's absorption fine structure (XAFS) were made at the Shanghai Light Source's Super XAS beamline (Shanghai, China). 2.9 T superbend magnet-provided incoming photon beam was chosen by a Si(111) channel-cut Quick-EXAFS monochromator. At 2.5 mrad, platinum-coated collimating and toroidal mirrors were used to focus and reject higher harmonics, respectively. Using a sample of Cu metallic powder, the beamline was calibrated. The portion of the material that the X-ray beam illuminated measured 0.5 mm by 0.2 mm. At room temperature, all spectra were captured in transmission mode. The 1 Hz frequency was used to obtain the EXAFS spectra, which were then averaged over a period of 5–15 min. By measuring the Cu pellet concurrently with each sample, the X-ray absorption near-edge structure (XANES) spectra were calibrated. Utilizing the Demeter software suite, the XAFS spectra were analyzed. Using a linear function, the background signal before the edge was removed (fitting range between −150 and −60 eV). After fitting it in the range between 150 and 1300–1450 eV after the edge, the post edge signal was scaled to a step of one. From the EXAFS fit of the Cu metal powder, we fitted the $k^3$ weighted Fourier transformed signal and calculated an amplitude reduction factor ($S_0^2$) of 0.78. The first coordination shell was fitted to all EXAFS spectra in the k-range of 0–15 $Å^{-1}$ and the R-range of 1–3 Å. To fit the Cu-Cu, Cu-O, and Cu-N scattering paths, Cu foil, CuO, $Cu_2O$, $CuCl_2$, and CuPc were used as the references.

Inductively coupled plasma-optical emission spectrometry (ICP-OES) was carried out on an Agient ICP-OES 5110 instrument. The sample (10 mg) was added into the digestion tank containing 7 mL mixed acid solution (V(HCl):V(HNO3) = 2:1) and 0.75 mL HF solution, followed by digestion treatment at 500 °C for several hours until the solid material had fully dissolved.

Laser Raman spectra were obtained on a LabRAM HR800 Evolutions (Horiba, Japan) spectrometer in the range of 400–3000 $cm^{-1}$ by employing an excitation wavelength of 532 nm line with Ar ion laser.

$N_2$ adsorption-desorption isotherms (BJH method) and Brunaur-Emmett-Teller (BET) surfaces areas were acquired with a surface area analyzer (BJ builder Kubo X1000).

A series of temperature-programmed tests were carried out by BJ builder PCA-1200 chemisorption analyzer, included $C_2H_2$- and HCl-TPD, In a typical experiment, about 100 mg sample was placed in a quartz reactor. $C_2H_2$, or HCl gas at the flow rate of 30 ml $min^{-1}$ was introduced to the quartz reactor at room temperature and a flow rate of 30 ml $min^{-1}$ for 1 h. Next, He (99.999%, for TPD) was used to purge the samples and pipeline for 1 h, followed by heating then the reactor was heated to 973 K (for TPD) at a constant heating rate of 5 K $min^{-1}$, while recording the signal of $C_2H_2$ (TPD-MS) and HCl (TPD-TCD) data.

A Linseis STA PT1600 system coupled to a Pfeiffer Vacuum Thermo-Star GSD 320 T1 mass spectrometer was used to perform the thermogravimetric analysis (TGA).

In-situ DRIFTS measurements were performed on a Nicolet 6700 FTIR spectrometer equipped with an in-situ high temperature and vacuum DRIFTS reaction cell (Harrick Praying Mantis). DRIFTS spectra were measured with 256 or 128 scans and a resolution of 2 $cm^{-1}$ using a MCT/A detector.

The production of •Cl was confirmed using the electron spin resonance (ESR) spin trapping technique on a Bruker A300 instrument, employing 5,5-dimethyl-1-pyrroline N-oxide (DMPO) as a trapping agent.

Aberration-corrected HAADF-STEM measurements were taken on a JEM-ARM200F instrument (University of Science and Technology of China) at 200 keV. Meanwhile, energy-dispersive X-ray elemental mapping was also collected on the same equipment. Transmission electron microscopy and STEM measurements were performed on a JEOL JEM2100F instrument operated at 200 keV.

High Energy Resolution Fluorescence Detected X-ray Absorption Spectroscopy (HERFD-XAS) was used to determine Cu speciation. Measurements were performed on FAME-UHD beamline at the European Synchrotron Radiation Facility (ESRF, Grenoble, France) at the Cu K-edge.

## Computational details

The Vienna Ab initio Simulation Package (VASP) code was used to perform DFT simulations[46–48]. Perdew-Becke-Ernzerhof was selected as the functional (PBE). Through the D3 approach, dispersion contributions were introduced[49,50]. Projector-augmented waves (PAW) were used to characterize the inner electrons, and plane waves with a cutoff kinetic energy of 450 eV were used to expand the valence monoelectronic states[51]. Three layers of a $(6 \times 5)$ supercell were used to model graphite slabs, interleaved by $15\,\text{Å}$ of vacuum, and relaxation was allowed only for the top two layers. The k-points sampling was a Γ-centered $3 \times 3 \times 1$ grid. The optimization thresholds for electronic and ionic relaxations were always $10^{-5}$ and $10^{-4}$ eV, respectively.

The reaction energy $(\Delta E)$ and activation $(E_a)$ energy were calculated by the Born-Oppenheimer energy difference between IS, TS, and FS:

$$\Delta E = E_{FS} - E_{IS} \tag{3}$$

$$E_a = E_{TS} - E_{IS} \tag{4}$$

where $E_{IS}$, $E_{TS}$, and $E_{FS}$ are the energies of IS, TS, and FS, respectively, and the equation for calculating $E_{ads}$ is given, as follows:

$$E_{ads} = E_{total} - E_{substrate} - E_{adsorbate} \tag{5}$$

where $E_{total}$, $E_{substrate}$, and $E_{adsorbate}$ are the total energies of the substrate and adsorbate, the energy of the substrate, and that of the adsorbate, respectively.

CDD was obtained by the following equation:

$$\Delta\rho = \rho_{AB} - \rho_A - \rho_B \tag{6}$$

where $\Delta\rho$ represents the charge of the total system, and ρA and ρB represent the charge of segment A and segment B, respectively.

The free energy of adsorbates was calculated from the harmonic vibrational energies $h\nu_i$ using VASPKIT (v.1.2.3) according to the following formulas:

$$F = U - T * S \tag{7}$$

$$U = E_{DFT} + E_{ZPE} + U(T) \tag{8}$$

$$S = \sum_i \left\{ \frac{\frac{h\nu_i}{kT}}{e^{\frac{h\nu_i}{kT}} - 1} - \ln\left[1 - e^{\frac{h\nu_i}{kT}}\right] \right\} \tag{9}$$

$$U(T) = \sum_i \frac{\frac{h\nu_i}{kT}}{e^{\frac{h\nu_i}{kT}} - 1} \tag{10}$$

where $F$, $U$, $T$, $S$, $E_{DFT}$, $E_{ZPE}$, $U(T)$ and vi represent free energy, internal energy, temperature, entropy, DFT energy, zero point energy, correction of heat capacity and vibrational frequencies, respectively. The **h** is Planck constant and **k** is Boltzmann constant.

## Data availability

The main data supporting the findings of this study are available within the article and its Supplementary Information. Additional data are available from the corresponding authors upon request. Source data are provided with this paper.

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

## Acknowledgements

Financial support from the National Key Research and Development Program of China (2021YFA1501801), Zhejiang Provincial Key R&D Project (2023C01211) and the National Natural Science Foundation of China (NSFC; grant nos. 22402179, 22472149) are gratefully acknowledged. We thank the EPSRC (EP/S018204/2, EP/Z001730/1 and EP/Y036220/1) for financial support. We thank the beamline scientists at the Diamond Light Source (I20 Scanning: SP38085-1), Soleil synchrotron (Galaxies beamline: 20221122) and the MAX IV (Balder beamline: 20230511) for the provision of beamtimes. The atom structures, chemical formulas and schematic illustrations were visualized using VESTA[52], BKChem, and Blender software.

## Author contributions

J.Z., F.W., Z.P. and X.L. supervised the work; Y.Y., M.Y., G.F. and B.W. designed the experiments, analyzed the results, and wrote the manuscript; Y.Y. and C.J. performed the DFT simulations; R.C. performed the in-situ DRIFTS; M.Y., S.W. and Y.Y. performed the catalysts preparation, catalytic tests and the TPD tests; T.S. and Y.Z. performed the aberration-corrected HAADF-STEM measurements and analyzed the results; Y.Y. and F.W. performed the XAFS experiments and analyzed the data; Z.Y. performed the HERFD-XANES experiments and analyzed the data.

## Competing interests

The authors declare no competing interests.
