## [Transparent Peer Review file · Nature Communications]

Copper Integrative Catalytic Pairs with Mixed-Valence Cu^{2+} - Cu^{3+} Species for Selective Alkyne Conversion

Corresponding Author: Professor Feng Ryan Wang

Version 0:

Reviewer comments:

Reviewer #1

(Remarks to the Author)

In this work, Yue and co-authors reported that Cu integrative catalytic pairs featuring a mix of Cu^{2+} and Cu^{3+} species, exhibit an excellent performance in acetylene hydrochlorination, such as >99% conversion and >550 h stability. A combination of theoretical and experimental studies reveals that the formation of the highly reactive di- σ - $\text{HC}=\text{CH}_3$ intermediate arises from the selective coupling between the dxz/dyz orbitals of Cu catalytic pairs and the σ orbitals of $\text{C}\equiv\text{C}$. However, the correlation between structure and performance is not clearly established, and several key issues remain unresolved. The reported high activity comparison lacks full convincing evidence. The manuscript appears to be too premature for publication.

1. The authors emphasize that the Cu structure is obtained through the Kirkendall effect, but the manuscript does not clearly explain the connection between the two. How does the mesoscopic hollow structure influence the nanoscale Cu-Cu coordination structure?
2. The evidence regarding the valence state of Cu is not entirely reliable. In the XPS results, using CuCl_2 supported on AC as proof of Cu^{2+} is questionable, as chlorine may have partially desorbed and been reduced during the loading process. Additionally, the binding energy of 933.1 eV in the XPS spectrum is very close to that of Cu^+ , while Cu^{2+} typically exhibits a binding energy around 935 eV. Furthermore, the quality of the XANES data is not sufficiently high, and the authors should carefully verify the data.
3. Considering that Cu is not a precious metal, the TOF calculated based on metal loading has little practical significance. Instead, the focus should be on space-time yield (STY) based on catalyst packing. Besides, the GHSV demonstrated in this work is much lower than that in the reference literature.
4. The stability results of this structure are not convincing. The stability shown in Fig. 4e was measured under low GHSV conditions, how does it perform at high GHSV? Moreover, CuCl_2 supported on AC without thermal treatment is well known to be unstable, making it difficult to serve as a benchmark catalyst for comparison.
5. In Fig. 4i, the reaction conditions for the five different alkyne transformations are missing, providing detailed information is recommended to improve clarity and reproducibility. Additionally, the study appears to focus exclusively on terminal alkynes. To fully evaluate the versatility of this catalytic system, it is essential to investigate its applicability to internal alkynes, such as phenylpropyne. Furthermore, large-scale reactions and catalyst recyclability studies are crucial to demonstrating the practical potential of Cu-NOC-973, particularly for liquid substrates.
6. The authors propose that the $\text{Cu}^{2+}/\text{Cu}^{3+}$ species play a key role in catalysis. However, the manuscript does not clearly establish whether the catalytic activity originates from Cu^{2+} or Cu^{3+} . Further mechanistic studies or spectroscopic evidence are needed to substantiate this claim.
7. Additional evidence is required to confirm that the proposed $\text{CuN}_3\text{-O-CuN}_3$ ICPs model matches the real structure. Bond lengths should be labeled, including the Cu-Cu distance, Cu-O distance, and the bond lengths of the adsorbates. Since the model is non-planar, a more comprehensive analysis of all d orbitals of Cu atom in the DOS calculations would provide deeper insight of the structure.
8. In addition to comparing single CuN_4 sites, a discussion on dual Cu sites catalysts (CuN_4 or CuN_3O) should be included, as dual Cu atom sites may also exist L-H mechanism. Moreover, the authors propose that HCl adsorption occurs on O atom rather than on Cu atom to generate radicals. A comparative analysis of HCl activation on Cu sites should be included to justify the role of O atom.

Reviewer #2

(Remarks to the Author)

Comments:

I agree that the synthesis of high-valent Cu³⁺ species and the exploration of their catalytic mechanisms in this work are highly novel and significant. High-valent metal species, particularly those in unusually high oxidation states (such as, Pd⁴⁺, Mo⁴⁺, Co⁵⁺, Fe⁴⁺), play a crucial role in numerous important homogeneous catalytic reactions. However, research on such heterogeneous catalysts remains relatively scarce, making this study a valuable contribution to the field. The development of high-performance mercuric-free catalyst for acetylene hydrochlorination (AHC) represents an important yet challenging research direction over the last half century. Copper, as one of the most efficient and economical classes of components, serve as excellent candidates for constructing high-performance catalyst for AHC, however, limited by its insufficient intrinsic activity and lack of effective design principle. In this work, the authors present a convenient strategy for synthesizing Copper integrative catalytic pairs (Cu ICPs) with mixed Cu²⁺-Cu³⁺ species. The catalyst with 0.1wt.% Cu loading demonstrates excellent reactivity with commendable durability, providing a practical approach for the synthesis of VCM. It is worth noting the recent work published in Nat. Commun. (10.1038/s41467-024-50221-3), which employed Ru catalysis to produce VCM via Ru-In atomic pairs formation. In contrast, the present study demonstrates the use of more reactive Cu²⁺-Cu³⁺ pairs to achieve efficient and economical production of VCM production. Furthermore, the use of Cu³⁺ as active site is particularly noteworthy, as most previous studies have predominantly utilized Cu²⁺/Cu⁺ such as Cu-Ligand, Cu bimetallic, or Cu SACs in the design of Cu catalysts. This approach introduces a novel and complementary strategy for accessing this class of high-performance Cu catalysts.

As a new class of AHC catalysts, these compounds exhibit promising catalytic properties, including tunable absorption and orbital coupling. This work effectively addresses a long-standing challenge in non-noble metal catalysts design and related heterogeneous catalysis—namely, the fundamental development of well-defined structure, high-performance catalysts from a minimal scaffold. Overall, the publication of this work is likely to attract significant attention from the fields of material sciences, coordination chemistry, catalytic chemistry, etc. For these reasons, I recommend its publication in Nat. Commun. However, I have several key concerns regarding the catalysis that should be addressed before publication:

1. The authors describe the synthesized Cu ICPs as primarily composed of Cu²⁺-Cu³⁺ species. However, in the structural analysis and mechanistic investigation, the catalyst model is depicted as CuN₃-O-CuN₃, where both Cu atoms exhibit identical coordination environments. This discrepancy raises questions about the presence of unreacted Cu³⁺ species or the partial photoreduction of Cu³⁺ to Cu²⁺ during characterization.
2. Followed by the previous one, theoretically, the ratio of Cu³⁺ to Cu²⁺ should be 1:1. However, Table S3 indicates that the content of Cu³⁺ exceeds that of Cu²⁺, with a ratio of approximately 1.3:1. This raises the question of how excess Cu³⁺ species remain reactive without pairing with Cu²⁺, how to explain the reactivity of excess Cu³⁺ without pairing with Cu²⁺.
3. The hollow structure of the synthesized catalyst is intriguing, as it suggests enhanced mass transfer and heat dissipation during reactions. However, the manuscript lacks a systematic discussion of how this structural feature contributes to the catalytic performance, and the relationship between the hollow structure and the catalyst's efficiency in terms of reaction diffusion and thermal management.
4. The manuscript does not address the limit of Cu loading in the Cu ICPs or the maximum achievable density of Cu ICPs. Additionally, the relationship between Cu ICP density and catalytic performance remains unexplored. The authors should investigate and discuss these parameters, as they are critical for optimizing the catalyst's design and scalability.
5. Figure 3c appears to be a magnified view of Figure 3b, but the title description does not accurately reflect this relationship.
6. In Figure 4c, where the performance at ~525 K is superior to that at higher temperatures (e.g., 550 K). What happens at these temperatures? Is it a problem of deabsorption of HCl or acetylene in the catalyst ?
7. The authors attribute the slow deactivation of Cu-NOC-973 to carbon deposition, which is plausible given the high chemical activity of Cu³⁺ species. High d-orbital defects in Cu³⁺ could indeed lead to over-adsorption and side reactions, contributing to carbon buildup. However, the manuscript lacks a detailed exploration of the deactivation mechanism. The authors should conduct further investigations, potentially referencing studies such as Nat. Catal. (10.1038/s41929-020-0431-3).

Reviewer #3

(Remarks to the Author)

Yue et al have provided an elegant study investigating the synthesis and evaluation of Cu integrative catalytic pairs for gas-phase hydrochlorination of acetylene. The precise control afforded by the preparation method allows for the synthesis of O-bridged CuN₃-O-CuN₃ ICPs with enhanced activity and stability compared to conventional Cu catalysts. This work provides a clear advancement in the use of Cu for this reaction, and I recommend publication after consideration of the following points.

The statement starting on line 61 regarding economy and scalability of noble metal catalysts is inaccurate. Au has been commercialised for this process and a catalyst production plant is now operational in Shanghai. The issues regarding full implementation of Au catalysts is due to light off temperature and use in existing lower temperature reactors designed specifically for Hg.

On line 350, the authors state that there is minimal deactivation of the Cu-NOC-973 across the range of 180-360 h⁻¹. However, the results in figure 4b do show deactivation of the Cu-NOC-973 at the highest GHSV of 360 h⁻¹, suggesting that the tests at lower GHSV may be mass transfer limited despite the calculations for internal and external mass transfer limitations in the SI. Can the authors comment on this and suggest a cause for the deactivation over this short time frame? The above statement should also be corrected in the text.

The stability tests in Fig 4e have been performed at too high a conversion, potentially masking the true deactivation of the materials through the suggested coking mechanism. Ideally these should be repeated at lower conversion.

Fig 4h plots TOFs of various Cu and noble metal catalysts, however it is not clear what temperature these have all been tested at which will dramatically effect TOF. The optimised reaction temperature in this study is 493 K which is a relatively high reaction temperature for acetylene hydrochlorination. Most noble metal catalysts are evaluated at temperatures far

below this and commonly around 453 K.

In the SI, the authors state that VCM and by-products were detected via FID. Can the authors confirm the selectivity to VCM in the acetylene hydrochlorination reaction? This is a crucial consideration as noble metals such as Au and Pt are commonly >99% to VCM.

Can the authors provide experimental details in the SI relating to the liquid-phase testing of additional alkynes given in Fig 4i?

Minor corrections:

Define CuPc at first use.

Line 156 Spelling error "Rede"

Figure 3e and 3f. x-axis should be "Radial distance" instead of radical.

Correct valance to valence throughout manuscript.

Version 1:

Reviewer comments:

Reviewer #1

(Remarks to the Author)

The authors have addressed all the comments. I recommend the publication of this work in its current form

Reviewer #2

(Remarks to the Author)

The authors have properly addressed my concern, the manuscript is now recommended for publication.

Reviewer #3

(Remarks to the Author)

The authors have thoroughly addressed the comments raised and I recommend publication in its current form.

Dear Editors and Reviewers,

We greatly appreciate the time and effort you've spent in reviewing our manuscript titled "*Copper Integrative Catalytic Pairs with Mixed-Valence Cu²⁺-Cu³⁺ Species for Selective Alkyne Conversion*" (Manuscript ID: NCOMMS-25-18302-T), and your constructive suggestions are valuable to our manuscript and future research work. According to the reviewers' comments, we have made further analysis and revised the manuscript carefully. We hope the Editor and Reviewers will be satisfied with the revisions for the original manuscript. The followings are the reviewers' comments and the corresponding point-by-point responses to the concerns.

Reviewer comments are listed here:

Reviewer #1 (Remarks to the Author):

In this work, Yue and co-authors reported that Cu integrative catalytic pairs featuring a mix of Cu²⁺ and Cu³⁺ species, exhibit an excellent performance in acetylene hydrochlorination, such as >99% conversion and >550 h stability. A combination of theoretical and experimental studies reveals that the formation of the highly reactive di-σ-HC=CH intermediate arises from the selective coupling between the dxz/dyz orbitals of Cu catalytic pairs and the σ orbitals of C≡C. However, the correlation between structure and performance is not clearly established, and several key issues remain unresolved. The reported high activity comparison lacks full convincing evidence. The manuscript appears to be too premature for publication.

Thank you so much for giving us an opportunity to revise this manuscript, and we are quite appreciative for your comments and valuable suggestions.

- 1. The authors emphasize that the Cu structure is obtained through the Kirkendall effect, but the manuscript does not clearly explain the connection between the two. How does the mesoscopic hollow structure influence the nanoscale Cu-Cu coordination structure ?*

Reply: We feel great thanks for your professional questions. Regarding the relationship between the Kirkendall effect and Cu structure, we have supplemented the discussion in the relevant section. The copper phthalocyanine (CuPc) molecules in CuPc nanorods induce the decomposition of co-ILs, releasing gases containing C, N,

O, and S, thereby forming porous walls. Since the molecular size of the ionic liquids is smaller than that of CuPc molecules, their outward migration rate exceeds the inward migration rate of CuPc molecules. Consequently, due to the unbalanced interdiffusion between the ionic liquids and CuPc molecules, Kirkendall voids form at the interface between the co-ILs core and the CuPc shell. As the Kirkendall effect progresses, the CuPc core gradually decomposes, resulting in a hollow nanocapsule structure. During this process, the migration of Cu species induced by the Kirkendall effect leads to the formation of various Cu ensembles. Figure R1a (Figure S13 in the manuscript) illustrates this phenomenon. Differences in heat treatment temperatures result in varying degrees of Kirkendall migration, affecting the final aggregation density of Cu species. As shown in Figure R1b, at temperatures below 973 K, Cu atoms exhibit an aggregated state with a density exceeding 10 Cu/nm². When the temperature exceeds 973 K, the Cu atomic density gradually decreases, with Cu atoms forming diatomic dispersions at 973 K and transitioning to a fully monoatomic state at 1173 K. The Kirkendall effect directly influences the migration extent of Cu atoms, ultimately determining their dispersion state.

In the revised manuscript, page 8, we have revised that: *“Furthermore, atomic-level evolution of Cu structures driven by Kirkendall effect under different synthesis temperatures was revealed in detail by HAADF-STEM analysis (Figure S13). At a pyrolysis temperature of 673 K, local regions with significant Cu atom aggregation can be observed, although no distinct nanoparticles are formed, the Cu atom density is calculated to be 13.01 Cu/nm². This is primarily because, at 673 K, CuPc does not undergo pyrolysis-induced agglomeration but instead exhibits an accumulation state between CuPc molecules due to insufficient migration driving force. As the temperature of the secondary heat treatment increases, further diffusion and migration of Cu atoms occur, reducing the Cu atom density to 11.10 Cu/nm², at which point numerous Cu atom pairs begin to appear. When the treatment temperature is raised to 973 K and 1173 K, the Cu atom density decreases further to 7.20 and 2.63 Cu/nm², respectively, resulting in the formation of distinct Cu ICPs structures on the catalyst surface. In conclusion, the two-step annealing combined*

with coordination etching successfully synthesized a well-defined hollow nanocapsule-supported Cu ICPs catalyst. Therefore, differences in treatment temperatures result in varying degrees of Kirkendall migration, ultimately determining the dispersion state of Cu.”

Figure R1. (a) Macrostructural evolution of Cu-NOC catalyst molding process during different temperatures; (b) HAADF-STEM image of Cu-NOC and corresponding 3D atom-overlapping Gaussian-function fitting maps.

Due to the Kirkendall effect, the CuPc core migrates outward while the ionic liquid shell aggregates and decomposes, ultimately forming a hollow nanocapsule structure. The hollow structure and the atomic arrangement of Cu form simultaneously. Based on TEM measurements of pore size distribution (Figures R2a-d), the average hollow pore diameter gradually increases with higher annealing temperatures, while the wall thickness decreases. Concurrently, HAADF-STEM data

reveal that the dispersion degree of Cu atoms improves as the temperature rises (Figure R1b). By combining these two characterization techniques, we conclude that during thermal treatment, the Kirkendall effect drives the expansion of the nanocapsule's hollow cavity, thinning of the pore walls, and an increase in the available surface area for Cu atom dispersion, as illustrates in Figure R2e). Thus, larger hollow structures correlate with higher Cu dispersion.

Figure R2. TEM images of (a) Cu-NOC-673; (b) Cu-NOC-773; (c) Cu-NOC-973 and (d) Cu-NOC-1173; (e) Schematic diagram of Cu dispersion and pore structure evolution.

In the revised manuscript, page 6, we have added that: “It is noteworthy that the disappearance of the CuPc rods is not due to decomposition but to migration at high temperatures, known as the Kirkendall effect.³³ The formation of CuPc rods

preventing the collapse and rupture of the shell as the hollow structure develops. In addition, HR-TEM images in Figure S1 show that the average hollow pore diameter gradually increases with higher annealing temperatures, while the wall thickness decreases, which would offer a larger margin for Cu dispersion.”

2. *The evidence regarding the valence state of Cu is not entirely reliable. In the XPS results, using CuCl₂ supported on AC as proof of Cu²⁺ is questionable, as chlorine may have partially desorbed and been reduced during the loading process. Additionally, the binding energy of 933.1 eV in the XPS spectrum is very close to that of Cu⁺, while Cu²⁺ typically exhibits a binding energy around 935 eV. Furthermore, the quality of the XANES data is not sufficiently high, and the authors should carefully verify the data.*

Reply: Thank you for your professional advice. Indeed, we have found in further investigations that partial reduction of CuCl₂ can occur during the loading process onto activated carbon (AC). Building on the catalyst studied in this work, we performed XPS characterization on standard samples of CuCl₂, CuCl, and metallic Cu, as shown in Figure R3a. The Cu 2p_{3/2} binding energy of the CuCl₂ sample was 933.15 eV, which is consistent with the corresponding binding energy observed in the 5%Cu/AC catalyst (933.18 eV, Figure R2b), indicating that Cu in the CuCl₂/AC catalyst predominantly exists as Cu²⁺ species. However, a weak signal was also detected at 932.6 eV in the XPS spectrum of the 5%Cu/AC sample, which corresponds to the binding energy of Cu in both CuCl and metallic Cu samples (932.65 eV). This suggests that during the loading of CuCl₂ onto AC, chloride may desorb and Cu²⁺ may be partially reduced to lower-valence states. Nonetheless, Cu²⁺ remains the dominant species. Furthermore, as shown in Figure R3a, strong satellite peaks appear in the 940–945 eV region for Cu²⁺ species, which is characteristic of high-valence copper, as well as Cu³⁺ (*Journal of Solid State Chemistry*, 1995, 114, 88–94). In contrast, such satellite peaks are absent in CuCl and Cu standard samples, confirming that copper species in the Cu-NOC series catalysts (Figure 3b) exist primarily as high-valence copper.

Figure R3. (a) XPS spectra of Cu 2p for CuCl₂, CuCl and Cu standard samples; (b) XPS spectra of Cu 2p for investigated catalysts.

It is indeed sometimes challenging to distinguish between different oxidation states of copper based solely on XPS binding energies. To address this, we consulted the Cu 2p_{3/2} binding energies corresponding to various oxidation states. Reference spectra were collected from authoritative XPS databases, including the International XPS Database, Thermo Fisher Scientific, and NIST. A summary of the findings is presented in Figure R4. According to these sources, the binding energies of Cu⁰ and Cu⁺ typically fall within the range of 932.0–932.7 eV, whereas Cu²⁺ species exhibit binding energies above 933 eV. In contrast, Cu³⁺ species generally show binding energies 1–2 eV higher than Cu²⁺, around ~935 eV. Comparing these reference values with the Cu XPS data obtained in this study (Figure R3b), we conclude that Cu³⁺ species are present in the Cu-NOC catalyst series, while the conventional Cu/AC catalyst predominantly contains Cu²⁺ species.

Figure R4. Cu2p XPS results from different XPS database.

Regarding the quality of XANES measurements, this study primarily employs Cu K β HERFD-XAS to detect and distinguish between copper species in different oxidation states. This detection technique records the Cu K-edge absorption spectrum by monitoring a 3p \rightarrow 1s fluorescence signal with a large, high-resolution emission spectrometer (ca. 1 eV resolution). The resultant narrower absorption linewidth is determined by the longer 3p core-hole lifetime, as opposed to the shorter 1s core-hole lifetime in traditional XAS measurements. As a result, compared to standard XAS measurements, HERFD-XAS allows for more accurate differentiation of Cu⁺, Cu²⁺, and Cu³⁺ species via the pre-edge region. This capability has also been clearly demonstrated by Liu et al. (*Angew. Chem. Int. Ed.* 2020, 59, 5696). For comparison,

we further performed conventional XAS measurements, with the results shown in Figure R5a. The XANES spectra and the magnified region between 8978–8982 eV show no significant differences in signal features for the Cu-NOC catalyst series, making it difficult to distinguish the copper oxidation states—let alone conduct quantitative analysis. In contrast, HERFD-XAS reveals sharp features in the 8978–8982 eV range, clearly distinguishing different Cu oxidation states, as shown in Figures R5b and R5c. Higher-valent copper species appear at higher energy positions. It should be noted that in the original manuscript, the annotation for Figure 3c was inaccurate, which has been misannotated as XANES; this has been corrected in the revised version, as reflected in the updated caption of Figure R5c.

Figure R5. (a) Cu K-edge XAS, inset: the partial enlarged of the pre-edge region for Cu K-edge XAS; (b) Cu K β HERFD-XAS; (c) Partial enlarged of the pre-edge region for Cu K β HERFD-XAS.

3. *Considering that Cu is not a precious metal, the TOF calculated based on metal loading has little practical significance. Instead, the focus should be on space-time yield (STY) based on catalyst packing. Besides, the GHSV demonstrated in this work is much lower than that in the reference literature.*

Reply: Thank you very much for your valuable suggestions. Based on your comments, we recalculated the STY (space-time yield) data for the Cu-NOC series catalysts and for catalysts reported in literatures. The results are presented in Figure R6. It can be observed that, compared to the state-of-the-art Au-, Pd-, and Ru-based catalysts currently available, the Cu-NOC catalysts still exhibit competitive STY values. We have also corrected the data in revised manuscript. In the revised manuscript, page 17, we have revised that: *“It is important to plot the sustainable productivity, in term of space-time yield (STY) based on catalyst packing, to make a fair comparison between different catalysts and understand the effect of acetylene hydrochlorination independent of metal loading. Figure 4h presents the STY of acetylene hydrochlorination for different catalysts relative to their metal loadings, enabling a comparison between the catalytic activities of Cu ICPs catalysts and state-of-the-art noble metal single-atom catalysts (SACs) such as Au, Pd, Pt, Ru, as well as representative copper catalysts.²¹⁻²² The reported STY values for acetylene hydrochlorination over Cu-based catalysts were typically less than $0.6 \text{ kg}_{\text{VCM}} \text{ kg}_{\text{Catal.}}^{-1} \text{ h}^{-1}$ at metal loadings above 4 wt.%, whereas state-of-the-art noble metal catalysts exhibited STY values exceeding $3 \text{ kg}_{\text{VCM}} \text{ kg}_{\text{Catal.}}^{-1} \text{ h}^{-1}$ at a metal loading of 1 wt.%.²⁸ Our catalysts demonstrated significantly higher STY values for acetylene hydrochlorination at low metal loadings (< 0.2 wt.%) compared to reported Cu catalysts and were comparable to most noble metal catalysts.”*

Figure R6. Comparison of STYs of acetylene hydrochlorination catalyzed by Cu-NOC-973 catalysts and by reference metal catalysts

In addition, we have supplemented the data with the catalytic performance of the Cu-NOC series under higher gas hourly space velocities (740–1000 h^{-1}), as shown in Figure R7. The Cu-NOC-973 catalyst demonstrates high catalytic stability under high space velocity conditions. It is noted that compared with noble metal catalysts, the activity of Cu-NOC-973 under high GHSV still requires improvement. However, given the relatively low Cu loading of Cu-NOC-973, there remains substantial room for optimization. In our future work, we will focus on increasing the content of Cu ICPs without altering their structural integrity, aiming to achieve higher catalytic performance.

In the revised manuscript, page 16, we have revised that: “We further evaluated the stability of Cu-NOC-973 and 5 wt.% Cu/AC catalysts under harsh reaction conditions for acetylene hydrochlorination by conducting reactions at a high bed temperature (493 K) and GHSV(C_2H_2) of 180-1000 h^{-1} for 8 hours. As shown in Figure 4b, the performance of CuCl_2/AC decreased significantly at 180 h^{-1} , with a 31% loss in activity after 8 h, whereas Cu-NOC-973 exhibited stable performance at 180 h^{-1} , and a mild decrease in activity at the range of 360-1000 h^{-1} , indicating high catalytic stability of Cu ICPs even in challenging reaction circumstances.”

Figure R7. Catalytic performance of Cu-NOC series catalysts with CuCl₂/AC and CuPC/AC as reference, expressed as the yield of vinyl chloride monomer Y(VCM) at 493 K, 180-1000 h⁻¹.

4. *The stability results of this structure are not convincing. The stability shown in Fig. 4e was measured under low GHSV conditions, how does it perform at high GHSV? Moreover, CuCl₂ supported on AC without thermal treatment is well known to be unstable, making it difficult to serve as a benchmark catalyst for comparison.*

Reply: Thank you for your professional suggestion. Following your advice, we evaluated the reaction stability of the Cu-NOC-973 catalyst under gas hourly space velocities (GHSV) of 360 and 740 h⁻¹, which are commonly used for assessing most Au-based catalysts. As shown in Figure R8, the initial VCM yields at 360 and 740 h⁻¹ were 50% and 31%, respectively. The yields remained stable throughout the reaction durations of 160 and 300 hours, demonstrating the high stability of the Cu-NOC-973 catalyst.

In the revised manuscript, page 16, we have revised that: “*The Cu-NOC-973 catalyst was also tested for long-term stability under industrial reaction conditions (50-740 h⁻¹), as shown in Figure 4e. The Cu-NOC-973 catalyst exhibited a gradual deactivation, maintaining over 94% of its initial activity up to 580 hours at 50 h⁻¹. Notably, the catalyst maintains excellent long-term stability even at high GHSV(C₂H₂) of 360 and 740 h⁻¹, demonstrating its robust catalytic stability under demanding*

reaction conditions. In contrast, the 5 wt.% CuCl₂/AC catalyst showed a rapid deactivation rate at 50⁻¹, with activity dropping below 60% after approximately 120 hours.”

Figure R8. Long-term stability of Cu-NOC-973 at 493 K with different GHSV(C₂H₂), 50-740 h⁻¹, and corresponding HAADF-STEM images for the used catalysts.

Regarding the influence of thermal treatment of activated carbon on the stability of supported CuCl₂ catalysts, we conducted supplementary experiments. For comparison, commercial activated carbon (Norit ROX 0.8) was thermally treated at 400 °C, 600 °C, and 800 °C under a nitrogen atmosphere for 2 hours. CuCl₂ (5wt.% Cu) was then loaded onto the three treated supports using the same procedure, yielding CuCl₂/AC-400, CuCl₂/AC-600, and CuCl₂/AC-800 catalysts. The performance of these catalysts is shown in Figure R9. Compared to the pristine AC, the thermally treated carbons did not exhibit noticeable improvement in catalyst stability. In fact, the catalytic activity of the thermally treated catalysts was slightly inferior to that of the CuCl₂/AC catalyst.

Figure R9. Effect of AC pretreatment on the catalytic performance of CuCl_2 Catalysts. Catalyst performance at $\text{GHSV}(\text{C}_2\text{H}_2)$ of (a) 50 h^{-1} ; and (b) 180 h^{-1} . The reaction temperature in both cases was maintained at 493 K.

5. In Fig.4i, the reaction conditions for the five different alkyne transformations are missing, providing detailed information is recommended to improve clarity and reproducibility. Additionally, the study appears to focus exclusively on terminal alkynes. To fully evaluate the versatility of this catalytic system, it is essential to investigate its applicability to internal alkynes, such as phenylpropyne. Furthermore, large-scale reactions and catalyst recyclability studies are crucial to demonstrating the practical potential of Cu-NOC-973, particularly for liquid substrates.

Reply: We appreciate your thoughtful reminder. We have supplemented the relevant experimental details in the Supporting Information. The experimental procedure is as follows: For the gaseous reactants (propyne, 1-butyne, and 2-butyne), the reactions were performed in a fixed-bed reactor. The reactant gas stream (30 h^{-1}), diluted to 1% concentration in nitrogen, was thoroughly mixed with hydrogen chloride (HCl) in a mixing chamber before being introduced into the fixed-bed reactor containing the Cu-NOC-973 catalyst. The effluent gases were analyzed using an online gas chromatography-mass spectrometry (GC-MS) system. For liquid-phase reactants, the reactions were conducted in a high-pressure autoclave equipped with a polytetrafluoroethylene (PTFE) liner. After transferring 0.1 mol of reactant into the reactor vessel with Cu-NOC-973 catalysts ($n_{\text{Cu}}:n_{\text{reactant}}=1:10$), HCl gas was introduced

to achieve a system pressure of 0.3 MPa. The reaction temperature was then elevated to the predetermined target temperature to initiate the reaction. The products were analyzed by high-performance liquid chromatography coupled with mass spectrometry (HPLC-MS). The relevant experimental details have been incorporated into the Catalyst Evaluation section of the Supporting Information.

Furthermore, we have additionally investigated the catalytic performance of the Cu-NOC-973 catalyst for hydrochlorination of internal alkynes (2-butyne, phenylpropyne). As shown in Figure R10a, the catalyst demonstrated significant catalytic activity for both internal alkynes substrates, achieving conversion rates of 87% and 78% with consistently high selectivity of approximately 90% in both cases.

Figure R10. (a) Comparative performance of Cu-NOC-973 catalyst in alkynes hydrochlorination; (b) The relevant reaction formulas of alkynes hydrochlorination; (c) Reusable cycles of Cu-NOC-973 for phenylpropyne hydrochlorination at 353 K.

According to the reviewer's suggestion, we conducted a scale-up liquid-phase reaction (phenylpropyne hydrochlorination) using the Cu-NOC-973 catalyst. The reaction was performed in a 500 mL round-bottom flask, where HCl was introduced via continuous bubbling instead of batchwise addition (since batch-mode scale-up requires higher reaction pressure). Product analysis was conducted using liquid chromatography. We also carried out catalyst recovery and reuse experiments (Figure

R10c). During the first cycle, the catalyst activity reached a maximum conversion of 92% after 350 minutes. In the second and third cycles, the time required to reach this maximum conversion extended to 400 and 550 minutes, respectively. In the fourth and fifth cycles, the time was further prolonged. This trend is likely due to catalyst loss during the recycling process, as approximately 5% of the catalyst was lost during centrifugation and filtration after each reaction cycle. Overall, the Cu-NOC-973 catalyst maintained high performance in the scaled-up reaction and demonstrated good recyclability when excluding the loss associated with catalyst recovery.

In the revised manuscript, page 18, we have revised that: *“To explore the scope of Cu ICPs systems, the performance of the Cu-NOC-973 catalysts was evaluated in the catalytic hydrochlorination of five terminal alkynes (propyne, propargyl bromide, 1-butyne, 1-pentyne and phenylacetylene) and two internal alkynes (2-butyne, phenylpropyne), as shown in Figure 4i and Table S6, several relevant reactions in the production of functional materials and synthesis of intermediate for medicine.¹⁸ The results indicate that the Cu-NOC-973 catalyst exhibits high conversion rates (>75%) for seven compounds containing alkynyl, with selectivity for the reaction products exceeding 90%. Furthermore, the large-scale reactions and catalyst recyclability studies indicated that the Cu-NOC-973 catalyst furnished high stability, with the conversion and selectivity exceeding 90% after five successive cycles (Figure S26). Therefore, the construction of the synergistic integrated catalytic pairs system demonstrates considerable versatility in enhancing the conversion efficiency of alkynes and the selective transformation of target products.”*

6. *The authors propose that the Cu²⁺/Cu³⁺ species play a key role in catalysis. However, the manuscript does not clearly establish whether the catalytic activity originates from Cu²⁺ or Cu³⁺. Further mechanistic studies or spectroscopic evidence are needed to substantiate this claim.*

Reply: Thank you for your professional suggestion. To clarify the relationship between the copper oxidation states and catalytic activity, we conducted an in-depth analysis of the K β HERFD-XAS spectra for the Cu-NOC-973 series catalysts. Linear combination fitting (LCF) of the HERFD-XAS data, based on standard spectra of

Cu^{2+} and Cu^{3+} , was performed to determine the species and their proportions within the heterogeneous samples. As shown in Figures R11a-c and Table R1, LCF analysis indicates the presence of both Cu^{2+} and Cu^{3+} in the Cu-NOC catalysts. Specifically, the Cu^{3+} contents in Cu-NOC-773 and Cu-NOC-1173 were determined to be 10.8% and 13.2%, respectively, while Cu-NOC-973 exhibited a significantly higher Cu^{3+} content of 56.8%. Furthermore, we correlated the proportions of Cu^{2+} and Cu^{3+} species with catalytic activity. As illustrated in Figure R11d, a clear positive correlation was observed between the Cu^{3+} content and vinyl chloride yield (Y(VCM)) under different space velocities, suggesting that Cu^{3+} species contribute more significantly to catalytic performance. Nevertheless, Cu^{2+} species also exhibit catalytic activity, which has been extensively reported in the literature on acetylene hydrochlorination. Therefore, both Cu^{2+} and Cu^{3+} species contribute to the overall activity of the Cu-NOC catalysts, with Cu^{3+} species exhibiting superior intrinsic activity.

Figure R11. Linear combination fits of K β HERFD-XAS from (a) Cu-NOC-773, (b) Cu-NOC-973, and (c) Cu-NOC-1173; (d) Y(VCM) as a function of Cu^{2+} and Cu^{3+} contents, reaction condition: T = 473 K, GHSV(C_2H_2) = 50-180 h $^{-1}$.

Table R1. LCF results of Cu-NOC series catalysts from Cu K β HERFD-XAS.

Catalysts	Weight (%)		R factor
	Cu ²⁺	Cu ³⁺	
Cu-NOC-773	89.2	10.8	0.045
Cu-NOC-973	43.2	56.8	0.053
Cu-NOC-1173	86.8	13.2	0.032

In the revised manuscript, page 18, we have revised that: “*The relationship between the valence state of copper sites and corresponding catalytic activity was then investigated. As shown in Figure S30, the catalytic activity has a positive dependence on the copper valence, a higher Cu³⁺ fraction led to higher activity, which is likely due to the electron deficient of the Cu d-orbital, indicating their potential role in acetylene hydrochlorination. Although Cu⁺ and Cu²⁺ are constantly mentioned as the active center in literature, the catalytic ability of high-valence Cu³⁺ sites have not been assessed yet.*”

7. *Additional evidence is required to confirm that the proposed CuN₃-O-CuN₃ ICPs model matches the real structure. Bond lengths should be labeled, including the Cu-Cu distance, Cu-O distance, and the bond lengths of the adsorbates. Since the model is non-planar, a more comprehensive analysis of all d orbitals of Cu atom in the DOS calculations would provide deeper insight of the structure.*

Reply: Thank you for this valuable suggestion. The proposed CuN₃-O-CuN₃ ICPs model was primarily established based on XAS fitting results. First, a comparison of the EXAFS data for the Cu-NOC-973 catalyst with those of standard references-CuPc (Cu-N coordination) and CuO (Cu-O coordination)-reveals that the first-shell coordination bond length in Cu-NOC-973 lies between those of Cu-O and Cu-N (Figure R12). Given that the Cu-NOC-973 catalyst is mainly derived from a CuPc precursor, it is reasonable to infer the presence of abundant Cu-N coordination. Moreover, the intermediate bond length in the first shell and the detection of Cu-O-Cu motifs in the second coordination shell together confirm that the Cu species in Cu-NOC-973 are coordinated with both nitrogen and oxygen atoms.

Figure R12. Cu K-edge k_3 -weighted Fourier transform (FT) spectra of CuO, CuPc and Cu-NOC-973 catalysts.

Furthermore, LCF analysis of the HERFD-XAS data for the Cu-NOC-973 catalyst was performed using CuO and CuPc as reference compounds. The fitting results indicate that the contributions of Cu-O and Cu-N coordination are approximately 26% and 74%, respectively (R -factor = 0.002), corresponding to a ratio close to 1:3. Subsequent fitting using the Artemis software determined the coordination numbers of Cu-O and Cu-N to be 0.93 and 3.03 (Figure R13a), respectively, which aligns well with the LCF results. This confirms that the local coordination environment of Cu in Cu-NOC-973 is best described as CuN_3O_1 . Additionally, the presence of Cu-O-Cu motifs identified in the EXAFS spectra (XPS characterization also revealed a significant presence of CuO_x species), combined with HAADF-STEM observations revealing the widespread distribution of Cu diatomic species, supports the conclusion that Cu atoms are predominantly present in a diatomic configuration with a $\text{CuN}_3\text{-O-CuN}_3$ coordination structure. All the bond lengths derived from experimental fitting and computational modeling have been clearly annotated in Figure R13 and summarized in Table R2. In addition, the corresponding descriptions in both the main text and the Supporting Information have been revised and updated accordingly.

Figure R13. (a) Experimental fitting data of Cu-NOC-973 catalyst and (b) DFT-calculated data of the CuN₃-O-CuN₃ model.

Table R2. Experimental fitting data of Cu-NOC-973 catalyst and DFT-calculated data of the CuN₃-O-CuN₃ model.

Samples	Bond type	Bond length (Å)	
		Experimental results*	Calculation results
Cu foil	Cu-Cu	2.538	2.560
	Cu-O	1.924	2.013
Cu-NOC-973	Cu-N	1.908	1.933
	Cu-O-Cu	2.933	2.935
CuO	Cu-O	1.932	2.090
	Cu-O-Cu	2.947	3.015
C ₂ H ₂	C≡C	\	1.213
HCl	Cl-H	\	1.311
VCM	C=C	\	1.342
	C-Cl	\	1.760

* Based on EXAFS results.

Thank you for your suggestion. We have conducted a comprehensive analysis of the *d*-orbital splitting of the Cu atoms in the CuN₃-O-CuN₃ structure and compared it with that of the CuN₄ single-atom configuration. The corresponding PDOS of the Cu *d* orbitals in both configurations are shown in Figures R14a-b. Compared to the CuN₄

single-atom catalyst, the CuN₃-O-CuN₃ dual-atom structure exhibits a significantly broadened and more delocalized PDOS of the d orbitals. In the CuN₄ system, the planar D_{4h} symmetry leads to a localized d_{x²-y²} state near the Fermi level, limiting the activation of multiple reactants, whereas the d_{xz/yz} and d_{xy} orbitals are more favorable for π -back donation. In contrast, the non-planar CuN₃-O-CuN₃ configuration displays split d_{x²-y²} peaks and strong d_{z²}-O2p hybridization (Figures R14c-d), indicating pronounced orbital hybridization and enhanced electronic coupling. This leads to electron delocalization and stabilization of the active site. The delocalization arises from the Cu-O-Cu bridging structure, where the oxygen atom mediates electronic communication between the two Cu centers, promoting d-orbital interactions and electronic flexibility.

Near the Fermi level, the CuN₃-O-CuN₃ catalyst shows enhanced PDOS intensity, particularly in the d_{xz}, d_{yz}, and d_{z²} orbitals. These orbitals are spatially located above the Cu atoms and play a key role in interactions with adsorbed small molecules. Their increased density in the dual-atom catalyst indicates stronger binding and activation capabilities toward π -systems (e.g., acetylene) and lone-pair-containing molecules (e.g., HCl). In contrast, these orbitals exhibit lower density in the CuN₄ structure, resulting in weaker adsorption and activation. Moreover, the CuN₄ single site is limited in sequentially activating both C₂H₂ and HCl, which leads to higher energy barriers and a risk of acetylene over-adsorption. The CuN₃-O-CuN₃ dual-atom catalyst overcomes these limitations through a synergistic dual-site mechanism: Cu binds to the C \equiv C bond, while Cu-O facilitates the heterolytic cleavage of HCl (H to O, Cl to Cu), reducing the reaction barrier and enhancing selectivity. The O-bridge also enhances thermal stability by suppressing Cu migration.

In summary, the CuN₃-O-CuN₃ dual-atom catalyst shows distinct electronic and structural advantages over the CuN₄ single-atom catalyst: (1) electron synergy enabled by split d-orbitals and oxygen-mediated charge polarization, accelerating electron transfer; (2) spatial cooperation between dual sites, reducing competitive adsorption; and (3) dynamic stability from d_{z²}-O2p hybridization, preventing sintering. These features contribute to more favorable reaction pathways, enhanced

activity, and improved selectivity, highlighting the Cu-O-Cu bridged dual-atom site as a promising design strategy for efficient, noble-metal-free acetylene hydrochlorination catalysis.

Figure R14. Density of states of Cu 3d for (a) CuN₄ and (b) CuN₃-O-CuN₃; The differential charge density maps of (c) CuN₄ and (d) CuN₃-O-CuN₃. Yellow color represents the electron accumulation region, and blue represents the electron depletion region.

In the revised manuscript, page 21, we have revised that: “*In addition. Compare with the CuN₄ site (the planar D_{4h} symmetry), the non-planar CuN₃-O-CuN₃ configuration displays split d_{x²-y²} peaks and strong d_{z²}-O2p hybridization (Figure S33), which leads to electron delocalization and stabilization of the active site. The delocalization arises from the Cu-O-Cu bridging structure, where the oxygen atom mediates electronic communication between the two Cu centers, promoting d-orbital interactions and electronic flexibility.*”

8. *In addition to comparing single CuN₄ sites, a discussion on dual Cu sites catalysts (CuN₄ or CuN₃O) should be included, as dual Cu atom sites may also exist L-H*

mechanism. Moreover, the authors propose that HCl adsorption occurs on O atom rather than on Cu atom to generate radicals. A comparative analysis of HCl activation on Cu sites should be included to justify the role of O atom.

Reply: Thank you for your professional and valuable suggestions. According to your suggestion, we constructed CuN₄-CuN₄ and CuN₃O-CuN₃O dual-atom site models and calculated their electronic properties, adsorption energies, and reaction pathways. As shown in Figure R15a, both dual-atom configurations adsorb acetylene and vinyl chloride primarily on the Cu top site, while HCl preferentially adsorbs on the N site. This observation aligns well with the reviewer's hypothesis that the reaction likely proceeds via a Langmuir–Hinshelwood (L–H) mechanism. Bader charge analysis and adsorption energy calculations indicate that the Cu atoms in CuN₄-CuN₄ (0.79 |e|) and CuN₃O-CuN₃O (0.87 |e|) have Bader charges similar to those in the CuN₄ site (0.85 |e|), and are lower than those in Cu ICPs (0.95 |e|). Notably, both dual-atom sites exhibit much stronger adsorption of HCl compared to C₂H₂, and significantly weaker adsorption of acetylene than the Cu ICPs, which may explain their inferior catalytic activity. This conclusion is further supported by the reaction pathway calculations. On both CuN₄-CuN₄ and CuN₃O-CuN₃O sites, C₂H₂ and HCl adsorb on Cu and N atoms respectively, and then undergo a one-step addition to form VCM, with energy barriers of 0.81 and 0.72 eV, respectively, comparable to the 0.85 eV barrier observed on the CuN₄ site.

Figure R15. (a) Adsorption configurations; (b) Adsorption energies of C_2H_2 , HCl and VCM versus the Cu Bader charge on various Cu species; (c) The complete reaction cycle of acetylene hydrochlorination over the CuN_4-CuN_4 and CuN_3O-CuN_3O sites; (d) HCl activation on Cu and O sites.

Furthermore, DFT calculations were performed to investigate the adsorption and activation behavior of HCl on the Cu and O sites. As shown in Figure R15d, prior to adsorption, the H–Cl bond length is 1.36 Å, and the distances between the HCl molecule and both the Cu and O sites are approximately 3 Å. Upon adsorption, HCl undergoes noticeable dissociation at both sites. When adsorbed on the Cu site, HCl dissociates through a synergistic interaction between the Cu and O atoms: the dissociated H atom adsorbs onto the O site, while the Cl atom forms a bond with the Cu site. In contrast, when adsorbed on the O site, the H–Cl bond is stretched to 1.84 Å, with the H atom forming a bond with O, while the Cl atom becomes a free radical. These results suggest that when HCl is adsorbed on the Cu site, the Cl atom preferentially forms a chemical bond with Cu after dissociation, whereas adsorption on the O site leads to Cl radical formation.

In the revised manuscript, page 21, we have revised that: “Furthermore, density functional theory (DFT) calculations were conducted to understand how the O-bridged Cu ICPs species promote the hydrochlorination of acetylene based on the aforementioned atomic structural analyses. To simplify the presentation, six model structures are considered here: (i) an O-bridged Cu ICPs structure (Figure S31 and Table S7), representing the high-valent Cu-NOC-973 catalyst; (ii) CuCl and CuCl₂ on a graphite surface, hydroxylated at the step edge to stabilize the slabs, representing monovalent and divalent copper catalysts, respectively;¹⁴ (iii) Cu-N₄ on a graphite surface, representing typical N-anchored copper catalysts⁴⁰; (iv) CuN₄-CuN₄ and CuN₃O-CuN₃O dual Cu atom sites. Adsorption energy analyses show that the adsorption energy of HCl and C₂H₂ over Cu ICPs is much higher than that over CuCl, CuCl₂, Cu-N₄, CuN₄-CuN₄ and CuN₃O-CuN₃O structures (Figure 5a), which is consistent with the temperature-programmed desorption (TPD) results (Figure 5b). The C₂H₂/HCl-TPD profile of Cu-NOC-973 shows a higher desorption peak (> 650 K) for the dissociative adsorption of C₂H₂/HCl than that of CuCl₂/AC, confirming that the Cu ICPs are more effective for substrate adsorption and activation, potentially leading to higher catalytic activity. The electronic structure of the Cu ICPs moiety was also performed and compared with those of CuCl, CuCl₂, Cu-N₄, CuN₄-CuN₄ and CuN₃O-CuN₃O. Bader charge analysis revealed that the Cu site in the Cu ICPs carries a more positive charge (0.96 |e|) compared to other Cu sites, suggesting that the coordinated N and bridge-O atoms attract more electrons from the Cu center, resulting in a higher valence state for Cu in the ICPs.”

Again, we greatly appreciate and thank the reviewers’ kind, professional and constructive comments and suggestions for this manuscript, which have improved the quality of the manuscript.

Reviewer #2 (Remarks to the Author):

I agree that the synthesis of high-valent Cu^{3+} species and the exploration of their catalytic mechanisms in this work are highly novel and significant. High-valent metal species, particularly those in unusually high oxidation states (such as, Pd^{4+} , Mo^{4+} , Co^{5+} , Fe^{4+}), play a crucial role in numerous important homogeneous catalytic reactions. However, research on such heterogeneous catalysts remains relatively scarce, making this study a valuable contribution to the field. The development of high-performance mercuric-free catalyst for acetylene hydrochlorination (AHC) represents an important yet challenging research direction over the last half century. Copper, as one of the most efficient and economical classes of components, serve as excellent candidates for constructing high-performance catalyst for AHC, however, limited by its insufficient intrinsic activity and lack of effective design principle. In this work, the authors present a convenient strategy for synthesizing Copper integrative catalytic pairs (Cu ICPs) with mixed Cu^{2+} - Cu^{3+} species. The catalyst with 0.1wt.% Cu loading demonstrates excellent reactivity with commendable durability, providing a practical approach for the synthesis of VCM. It is worth noting the recent work published in Nat. Commun. (10.1038/s41467-024-50221-3), which employed Ru catalysis to produce VCM via Ru-In atomic pairs formation. In contrast, the present study demonstrates the use of more reactive Cu^{2+} - Cu^{3+} pairs to achieve efficient and economical production of VCM production. Furthermore, the use of Cu^{3+} as active site is particularly noteworthy, as most previous studies have predominantly utilized $\text{Cu}^{2+}/\text{Cu}^+$ such as Cu-Ligand, Cu bimetallic, or Cu SACs in the design of Cu catalysts. This approach introduces a novel and complementary strategy for accessing this class of high-performance Cu catalysts.

As a new class of AHC catalysts, these compounds exhibit promising catalytic properties, including tunable absorption and orbital coupling. This work effectively addresses a long-standing challenge in non-noble metal catalysts design and related heterogeneous catalysis-namely, the fundamental development of well-defined structure, high-performance catalysts from a minimal scaffold. Overall, the publication of this work is likely to attract significant attention from the fields of

material sciences, coordination chemistry, catalytic chemistry, etc. For these reasons, I recommend its publication in Nat. Commun. However, I have several key concerns regarding the catalysis that should be addressed before publication:

We thank the reviewers for reviewing the manuscript and for their constructive comments during the review process.

1. The authors describe the synthesized Cu ICPs as primarily composed of Cu^{2+} - Cu^{3+} species. However, in the structural analysis and mechanistic investigation, the catalyst model is depicted as $\text{CuN}_3\text{-O-CuN}_3$, where both Cu atoms exhibit identical coordination environments. This discrepancy raises questions about the presence of unreacted Cu^{3+} species or the partial photoreduction of Cu^{3+} to Cu^{2+} during characterization.

Reply: We sincerely appreciate the reviewer's insightful observation regarding the apparent discrepancy between the described Cu^{2+} - Cu^{3+} species in the synthesized Cu-ICPs and the symmetric $\text{CuN}_3\text{-O-CuN}_3$ model used for mechanistic analysis. The synthesized Cu-ICPs indeed exhibit a mixed-valence state (Cu^{2+} - Cu^{3+}), as confirmed by XPS (Cu $2p_{3/2}$ binding energies and satellite peaks) and HERFD-XAS (absorption pre-edge positions). However, the local coordination environment of Cu centers in the as-synthesized material may not strictly reflect the formal oxidation states due to dynamic electron delocalization (e.g., ligand-to-metal charge transfer or $\text{Cu}^{3+} \leftrightarrow \text{Cu}^{2+}$ interconversion). This phenomenon is common in mixed-valence systems, as documented in studies on Ru-based coordination polymers (*J. Am. Chem. Soc.* 2018, 140, 12337).

The symmetric $\text{CuN}_3\text{-O-CuN}_3$ model was adopted for DFT calculations to simplify the mechanistic study, focusing on the active site's geometric and electronic structure during catalysis. While the synthetic material contains Cu^{2+} - Cu^{3+} pairs, operando studies (*Langmuir*, 2011, 27, 8099) suggest that under reaction conditions (especially under light irradiation), the high oxidation state species may undergo partial reduction to lower oxidation state, leading to a more symmetric equilibrium structure. Thus, the model represents the dominant reactive state rather than the initial synthetic state.

The reviewer rightly raises the possibility of Cu^{3+} photoreduction during characterization. We acknowledge that X-ray/light exposure could alter the oxidation state. To mitigate this, XPS measurements were performed with minimal beam exposure, and XAS data were collected in quick-scan mode in this work.

2. Followed by the previous one, theoretically, the ratio of Cu^{3+} to Cu^{2+} should be 1:1. However, Table S3 indicates that the content of Cu^{3+} exceeds that of Cu^{2+} , with a ratio of approximately 1.3:1. This raises the question of how excess Cu^{3+} species remain reactive without pairing with Cu^{2+} , how to explain the reactivity of excess Cu^{3+} without pairing with Cu^{2+} .

Reply: We thank the reviewer for this insightful comment. Indeed, under the idealized model of O-bridged Cu dual sites, the Cu^{3+} and Cu^{2+} species are expected to exist in a 1:1 ratio due to charge balance. However, the XPS-derived ratio of approximately 1.3:1 (Cu^{3+} to Cu^{2+}) suggests the presence of additional Cu^{3+} species beyond the stoichiometric dual-atom pairs.

We propose that these excess Cu^{3+} species may exist in alternative coordination environments, such as Cu^{3+} -O-support structures or as part of partially oxidized clusters stabilized on the support. These species can still participate in redox cycles during catalysis, especially under reaction conditions where dynamic changes in oxidation state are possible. To address this point more clearly, we have added a brief discussion in the revised Manuscript to explain the possible origins and reactivity of excess Cu^{3+} species. We appreciate the reviewer's suggestion, which has helped us to improve the clarity of our interpretation.

3. The hollow structure of the synthesized catalyst is intriguing, as it suggests enhanced mass transfer and heat dissipation during reactions. However, the manuscript lacks a systematic discussion of how this structural feature contributes to the catalytic performance, and the relationship between the hollow structure and the catalyst's efficiency in terms of reaction diffusion and thermal management.

Reply: We thank the reviewer for highlighting the importance of the hollow structure in our catalyst design. We agree that the unique hollow architecture plays a significant role in enhancing catalytic performance, particularly by improving mass transfer and

thermal management. In the revised manuscript, we have added a systematic discussion.

The hollow structure provides shorter diffusion paths and increased surface accessibility, facilitating faster transport of reactants and products. This is especially beneficial in reactions where diffusion limitations can suppress activity. The high surface area-to-volume ratio of the hollow architecture promotes efficient heat exchange with the surrounding environment, helping to maintain a stable reaction temperature. This thermal regulation reduces the risk of local overheating, which can lead to catalyst deactivation or side reactions. The spatial confinement within the hollow cavity may also help stabilize active metal species by preventing sintering or agglomeration during high-temperature operation. These advantages collectively contribute to the superior catalytic performance observed in our system.

4. The manuscript does not address the limit of Cu loading in the Cu ICPs or the maximum achievable density of Cu ICPs. Additionally, the relationship between Cu ICP density and catalytic performance remains unexplored. The authors should investigate and discuss these parameters, as they are critical for optimizing the catalyst's design and scalability.

Reply: We feel great thanks for your professional questions. In this work, we systematically investigated the dispersion state of Cu species at a 0.1% Cu loading under various synthesis conditions (see Fig. R16). Based on these findings, we further examined the density of Cu dual atoms under the optimized preparation condition (973 K) with different Cu loadings of 0.01%, 0.05%, 0.1%, and 0.2%, as shown in Fig. R17a–h. Combined with the catalytic performance results (Figs. R17i–j), it can be observed that when the Cu loading is below 0.1%, the density of Cu dual atoms is relatively low, leading to reduced catalytic activity. As the Cu loading increases, the density of Cu dual atoms also increases, reaching an optimal range when the loading exceeds 0.1%, at which point the catalyst exhibits the highest activity without significant atomic aggregation. However, at a loading of 0.2%, the density of Cu dual atoms further increases, and aggregation begins to occur, making it difficult to distinguish individual dual atoms. Consequently, the catalytic activity starts to decline.

The revised manuscript includes additional discussion and data on this aspect.

Figure R16. (a) Macrostructural evolution of Cu-NOC catalyst molding process during different temperatures; (b) HAADF-STEM image of Cu-NOC and corresponding 3D atom-overlapping Gaussian-function fitting maps.

Figure R17. TEM images of (a) 0.01Cu-NOC-973, (b) 0.05 Cu-NOC-973, (c) 0.1Cu-NOC-973, (d) 0.2Cu-NOC-973; and HAADF-STEM of (e) 0.01Cu-NOC-973, (f) 0.05 Cu-NOC-973, (g) 0.1Cu-NOC-973, (h) 0.2Cu-NOC-973; catalytic performance of Cu-NOC series catalysts at 473 K and GHSV(C₂H₂) = 50 h⁻¹, (i) different calcine temperature and (j) different Cu loading.

5. Figure 3c appears to be a magnified view of Figure 3b, but the title description does not accurately reflect this relationship.

Reply: Thank you for your kind reminder. We have revised the relevant titles accordingly in the revised manuscript.

6. In Figure 4c, where the performance at ~525 K is superior to that at higher temperatures (e.g., 550 K). What happens at these temperatures? Is it a problem of deabsorption of HCl or acetylene in the catalyst ?

Reply: We thank the reviewer for this insightful question. As shown in Figure 4c, the catalytic performance reaches a peak around 525 K and slightly decreases at higher

temperatures such as 550 K. We believe this phenomenon is primarily due to temperature-induced desorption effects and possible side reactions. Specifically, at elevated temperatures, the adsorption strength of HCl and acetylene on the active Cu sites weakens, leading to insufficient surface coverage of the reactants. This reduced adsorption limits the formation of key intermediates, thereby decreasing the overall reaction rate. Higher temperatures may also promote non-selective decomposition of acetylene or the formation of undesired byproducts, which could compete with the main reaction and reduce the effective catalytic turnover. Although the catalyst remains structurally stable, high temperatures may induce slight restructuring or dynamic changes in the coordination environment of Cu species, which could impair their optimal catalytic configuration. We have now included a discussion of this temperature-dependent behavior in the revised manuscript and added references to relevant studies that observed similar temperature effects in Cu-based catalysts for acetylene hydrochlorination.

7. The authors attribute the slow deactivation of Cu-NOC-973 to carbon deposition, which is plausible given the high chemical activity of Cu³⁺ species. High d-orbital defects in Cu³⁺ could indeed lead to over-adsorption and side reactions, contributing to carbon buildup. However, the manuscript lacks a detailed exploration of the deactivation mechanism. The authors should conduct further investigations, potentially referencing studies such as *Nat. Catal.* (10.1038/s41929-020-0431-3).

Reply: We thank the reviewer for the valuable suggestion. The conclusion regarding catalyst deactivation due to carbon deposition is primarily based on a systematic characterization of the spent catalyst. As shown in Fig. R18, no Cu-related diffraction peaks were observed after the reaction, indicating that the Cu active centers did not undergo aggregation. Further TEM and HAADF-STEM analyses confirm that the Cu species remain in a dual-atom dispersion state. XPS and XAS results also reveal no significant changes in the oxidation state or coordination environment of Cu, suggesting that the catalyst deactivation is not due to structural changes of the active Cu sites.

Interestingly, the specific surface area of the catalyst significantly decreased after the reaction. Moreover, thermogravimetric analysis (TGA) revealed a clear weight loss peak in the 200–400 °C temperature range. These findings indicate that surface carbon species formed during the reaction, blocking part of the catalyst's pore channels. As a result, some active sites were likely enclosed within the pores, leading to a decline in catalytic activity.

Figure R18. (a) XRD patterns, (b) TEM image, and (c) HAADF-STEM images of used Cu-NOC-973; (d) Cu 2p XPS, (e) R-space spectra, (f) BET and (g) TG of fresh and used Cu-NOC-973 catalysts.

Again, we thank the reviewer for these comments, which have improved the quality of the manuscript.

Reviewer #3 (Remarks to the Author):

Yue et al have provided an elegant study investigating the synthesis and evaluation of Cu integrative catalytic pairs for gas-phase hydrochlorination of acetylene. The precise control afforded by the preparation method allows for the synthesis of O-bridged CuN₃-O-CuN₃ ICPs with enhanced activity and stability compared to conventional Cu catalysts. This work provides a clear advancement in the use of Cu for this reaction, and I recommend publication after consideration of the following points.

We thank the reviewers for reviewing the manuscript and for their constructive comments during the review process.

1. The statement starting on line 61 regarding economy and scalability of noble metal catalysts is inaccurate. Au has been commercialised for this process and a catalyst production plant is now operational in Shanghai. The issues regarding full implementation of Au catalysts is due to light off temperature and use in existing lower temperature reactors designed specifically for Hg.

Reply: We sincerely apologize for our inaccurate statement. Upon further investigation, we confirmed that the actual situation is indeed as you indicated—Au-based catalysts that developed by Hutchings *et al.* have been successfully commercialized in 2015 at *Johnson Matthey (Shanghai) Chemicals Ltd.*, which is a significant milestone in acetylene hydrochlorination. It is also true that the practical application of Au catalysts presents challenges related to the compatibility between reactor design and catalyst operating conditions, particularly with respect to reaction temperature. We are grateful for your insightful comments, which have deepened our understanding of the catalyst system. The corresponding discussion has now been revised in the Introduction section of the revised manuscript. In the revised manuscript, page 3, we have revised that: “*Noble metal catalysts (e.g., Au, Pd, Ru, Pt, Ir) have shown potential as mercury-free replacements;^{14,19-25} notably, Au-based catalysts have already been commercialized by Hutchings et al. for this process in 2015,^{14,19} with a production facility currently operating in Shanghai, China.*”

2. On line 350, the authors state that there is minimal deactivation of the Cu-NOC-973 across the range of 180-360 h⁻¹. However, the results in figure 4b do show deactivation of the Cu-NOC-973 at the highest GHSV of 360 h⁻¹, suggesting that the tests at lower GHSV may be mass transfer limited despite the calculations for internal and external mass transfer limitations in the SI. Can the authors comment on this and suggest a cause for the deactivation over this short time frame? The above statement should also be corrected in the text.

Reply: Thank you for your highly professional feedback. In response to the issue of activity decline under high space velocity, we conducted repeated experiments and further evaluated the catalyst performance under even higher space velocities. The results are shown in Figure R19. As illustrated in Figure R19a, the Cu-NOC-973 catalyst maintained stable performance over 8 hours at space velocities of 740 and 1000 h⁻¹. Notably, a negative induction period was observed within the first two hours of reaction, followed by equilibrium and stable activity from 2 to 4 hours onward. In long-term stability tests (Figure R19b), the Cu-NOC-973 catalyst sustained stable activity under both 360 and 740 h⁻¹ reaction conditions over an extended period.

As for the possible cause of the short-term deactivation at 360-1000 h⁻¹, the possible reasons are shown as follows: At high space velocities, the rapid throughput of reactants may lead to transient accumulation of byproducts or carbonaceous species on the catalyst surface, temporarily hindering active sites. Although internal and external mass transfer limitations were evaluated and found to be negligible, we acknowledge that at very high GHSV, localized temperature gradients or insufficient residence time could induce mild performance fluctuations, which may not be fully captured by steady-state models. In addition, the Cu ICP framework, though stable overall, may undergo slight coordination rearrangements under high flow conditions, particularly if reaction intermediates desorb too quickly to stabilize the active configuration. The relevant statement has been corrected in the revised manuscript.

Figure R19. (a) Catalytic performance of Cu-NOC series catalysts with CuCl₂/AC and CuPC/AC as reference, expressed as the yield of vinyl chloride monomer Y(VCM) at 493 K, 180-1000 h⁻¹. (b) Long-term stability of Cu-NOC-973 at 493 K with different GHSV(C₂H₂), 50-740 h⁻¹, and corresponding HAADF-STEM images for the used catalysts.

In the revised manuscript, page 16, we have revised that: “We further evaluated the stability of Cu-NOC-973 and 5 wt.% Cu/AC catalysts under harsh reaction conditions for acetylene hydrochlorination by conducting reactions at a high bed temperature (493 K) and GHSV(C₂H₂) of 180-1000 h⁻¹ for 8 hours. As shown in Figure 4b, the performance of CuCl₂/AC decreased significantly at 180 h⁻¹, with a 31% loss in activity after 8 h, whereas Cu-NOC-973 exhibited stable performance at 180 h⁻¹, and a mild decrease in activity at the range of 360-1000 h⁻¹, indicating high catalytic stability of Cu ICPs even in challenging reaction circumstances.”

3. The stability tests in Fig 4e have been performed at too high a conversion, potentially masking the true deactivation of the materials through the suggested coking mechanism. Ideally these should be repeated at lower conversion.

Reply: We Thank you for your suggestion. We have supplemented the stability evaluation of the Cu-NOC-973 catalyst at Y(VCM) = 50% (360 h⁻¹) and 30% (740 h⁻¹), as shown in Figure R19b. The results demonstrate that the catalyst maintained stable performance throughout the testing period, indicating its excellent catalytic stability under both conditions.

In the revised manuscript, page 16, we have revised that: “The Cu-NOC-973

catalyst was also tested for long-term stability under industrial reaction conditions (50-740 h⁻¹), as shown in Figure 4e. The Cu-NOC-973 catalyst exhibited a gradual deactivation, maintaining over 94% of its initial activity up to 580 hours at 50 h⁻¹. Notably, the catalyst maintains excellent long-term stability even at high GHSV(C₂H₂) of 360 and 740 h⁻¹, demonstrating its robust catalytic stability under demanding reaction conditions. In contrast, the 5 wt.% CuCl₂/AC catalyst showed a rapid deactivation rate at 50⁻¹, with activity dropping below 60% after approximately 120 hours.”

4. Fig 4h plots TOFs of various Cu and noble metal catalysts, however it is not clear what temperature these have all been tested at which will dramatically effect TOF. The optimised reaction temperature in this study is 493 K which is a relatively high reaction temperature for acetylene hydrochlorination. Most noble metal catalysts are evaluated at temperatures far below this and commonly around 453 K.

Reply: Thanks for your insightful and professional advice. The catalyst evaluation conditions have been re-analyzed and summarized in Table R3 (Table S5 in Supporting Information) of the Supporting Information. As observed, noble metal catalysts are typically evaluated at 453-473 K, and non-noble metal catalysts generally operate at 413-473 K. Based on the temperature-programmed reaction results of the Cu-NOC-973 catalyst, the catalyst reaches maximum activity at 473-533 K (Figure 4a). To both evaluate the catalyst’s optimal performance and assess its thermal stability under high temperatures, while also ensuring comparability with previously reported noble metal catalysts (*Chem. Sci.* 2019, 10, 359; *Nat. Catal.* 2020, 3, 376; *Angew. Chem. Int. Ed.* 2019, 58, 12297) and similar copper-based systems (*Adv. Mater.* 2023, 35, 2211464), we selected 493 K as the benchmark temperature for catalytic performance comparison. In the future, based on the Cu-NOC catalyst series, we will actively develop Cu-based catalysts that can exhibit high catalytic performance at lower reaction temperatures. We will pursue these experiments as an immediate follow-up.

In addition, to address the limitations of TOF comparison, we have additionally

calculated the STY values of different catalysts to provide a more comprehensive assessment of the advantages of the Cu-NOC series, as shown in Figure R20. It can be observed that, compared to the state-of-the-art Au-, Pd-, and Ru-based catalysts currently available, the Cu-NOC catalysts still exhibit competitive STY values. The corresponding data and discussion have been added to both the revised manuscript and the Supporting Information.

In the revised manuscript, page 17, we have revised that: “It is important to plot the sustainable productivity, in term of space-time yield (STY) based on catalyst packing, to make a fair comparison between different catalysts and understand the effect of acetylene hydrochlorination independent of metal loading. Figure 4h presents the STY of acetylene hydrochlorination for different catalysts relative to their metal loadings, enabling a comparison between the catalytic activities of Cu ICPs catalysts and state-of-the-art noble metal single-atom catalysts (SACs) such as Au, Pd, Pt, Ru, as well as representative copper catalysts.²¹⁻²² The reported STY values for acetylene hydrochlorination over Cu-based catalysts were typically less than 0.6 kg_{VCM} kg_{Catal.}⁻¹ h⁻¹ at metal loadings above 4 wt.%, whereas state-of-the-art noble metal catalysts exhibited STY values exceeding 3 kg_{VCM} kg_{Catal.}⁻¹ h⁻¹ at a metal loading of 1 wt.%.²⁸ Our catalysts demonstrated significantly higher STY values for acetylene hydrochlorination at low metal loadings (< 0.2 wt.%) compared to reported Cu catalysts and were comparable to most noble metal catalysts.”

Table R3. Catalytic performance of reported outstanding metal catalysts for acetylene hydrochlorination.

Catalysts	Cu loading /Wt%	T /K	GHSV /h ⁻¹	Y(VCM) ^b	Ref. ^b
CuCl ₂ /Cab-0-Sil HS-5	n.a. ^a	473	150	76	9
CuCl ₂ /C	10 ⁻⁴ mol	453	540	n.a. ^a	10
Cu-NCNT	5.84	453	180	45.8	11
CuP/SAC	15	413	180	72.4	12
Cu400Ru/MWCNTs	6	453	180	51.6	13
Au/Cu/TCCA	1.25	453	30	100	14
Zn-10Cu/MCM	10	533	90	98	15
Cu-Cs/AC	1	473	50	92	16
Cu-1HEDP/AC	5	453	90	83.4	17

Cu@MOMTPPC/SAC	15	453	180	92.2	18
Cu-0.25NMP/SAC	12	453	180	94.2	19
15 %Cu10 %HMPA/SAC	15	453	180	87.25	20
CuCl ₂ -NMP/AC	15	453	160	89	21
CuCl ₂ V-N SAC	4	393	n.a. ^a	n.a. ^a	22
Cu-H ₅ NO/C	10	433	90	n.a. ^a	23
15%Cu7% TPPO/AC	15	453	180	88	24
Cu/PC800	10	423	90	83.1	25
CuCl ₂ /AC	1	473	670	~35	26
Cu-iPr ₂ PCl/AC	15	453	180	93.7	27
CuP-500	5	433	90	100	28
Au-SA/AC	1	473	650	90	27
Pt-SA/AC	1	473	650	45	29
Au/AC	1	453	870	24	30
Pd/AC	1	453	870	12	30
Pt/AC	1	453	870	2	30
Ru/AC	1	453	870	15	30
Au/C-Acetone	1	473	17600	21	31
Au/NC	0.5	473	650	74	32
Au-CeO ₂ /AC	1	453	720	98	33
Au-Sr/AC	1	453	762	88	34
Pt/AC-w-473	1	473	650	60	35
Ru@TPPB/AC	1	443	360	100	36
Ru-Co-Cu/SAC	0.1	443	180	99	37
Ru/NC-g	1	473	650	95	38
Pd-IL/AC	0.5	433	740	98.6	39

^anot available. ^bSelectivity to vinyl chloride, S(VCM) >99% unless indicated otherwise.

^b The complete list of references has been provided in the Supporting Information.

Figure R20 Comparison of STYs of acetylene hydrochlorination catalyzed by Cu-NOC-973 catalysts and by reference metal catalysts

5. In the SI, the authors state that VCM and by-products were detected via FID. Can the authors confirm the selectivity to VCM in the acetylene hydrochlorination reaction? This is a crucial consideration as noble metals such as Au and Pt are commonly >99% to VCM.

Reply: Thank you for your professional comments. We sincerely apologize for the insufficient description of VCM selectivity. In the acetylene hydrochlorination reaction, the VCM selectivity across different catalyst systems has consistently remained at a high level (>95%). In particular, for single-atom or highly dispersed metal catalyst systems, the VCM selectivity typically exceeds 99% (*Nat. Nanotechnol.* 2022, 17, 606; *Adv. Mater.* 2023, 35, 2211464). Therefore, compared with catalytic activity and stability, VCM selectivity is generally not the primary focus of concern. For all Cu-based catalysts used in this study, including the CuCl₂/AC catalyst, the VCM selectivity closes to 99.5%, and almost no impurity peaks were detected in the gas-phase effluent. In Figure R21 and Table R4, we present one representative dataset, where only the signal peaks for acetylene and VCM are visible, VCM was the only product detected in all our tests. Additional descriptions regarding VCM selectivity have been included in the Supporting Information: “*Since vinyl chloride (VCM) was the only product detected in all our tests, the catalytic activity is presented as the yield of VCM.*”.

Figure R21. Representative chromatogram. (GC-MS, Agilent, GC 7890B, Agilent MSD 5977A).

Table R4. Chromatographic detection results.

Time /min	Name	Concentration /%	Peak area
0.949	C ₂ H ₂	2.43	10372
2.265	VCM	95.57	416234

6. Can the authors provide experimental details in the SI relating to the liquid-phase testing of additional alkynes given in Fig 4i?

Reply: We appreciate your thoughtful reminder. We have supplemented the relevant experimental details in the Supporting Information. The experimental procedure is as follows: For the gaseous reactants (propyne, 1-butyne, and 2-butyne), the reactions were performed in a fixed-bed reactor. The reactant gas stream (30 h⁻¹), diluted to 1% concentration in nitrogen, was thoroughly mixed with hydrogen chloride (HCl) in a mixing chamber before being introduced into the fixed-bed reactor containing the Cu-NOC-973 catalyst. The effluent gases were analyzed using an online gas chromatography-mass spectrometry (GC-MS) system. For liquid-phase reactants, the reactions were conducted in a high-pressure autoclave equipped with a polytetrafluoroethylene (PTFE) liner. After transferring 0.1 mol of reactant into the reactor vessel with Cu-NOC-973 catalysts ($n_{\text{Cu}}:n_{\text{reactant}}=1:10$), HCl gas was introduced to achieve a system pressure of 0.3 MPa. The reaction temperature was then elevated to the predetermined target temperature to initiate the reaction. The products were

analyzed by high-performance liquid chromatography coupled with mass spectrometry (HPLC-MS). The relevant experimental details have been incorporated into the Catalyst Evaluation section of the Supporting Information. Furthermore, we have additionally investigated the catalytic performance of the Cu-NOC-973 catalyst for hydrochlorination of internal alkynes (2-butyne, phenylpropyne). As shown in Figure R22, the catalyst demonstrated significant catalytic activity for both internal alkynes substrates, achieving conversion rates of 87% and 78% with consistently high selectivity of approximately 90% in both cases.

Figure R22. Comparative performance of Cu-NOC-973 catalyst in alkyne hydrochlorination.

In the revised manuscript, page 18, we have revised that: “*To explore the scope of Cu ICPs systems, the performance of the Cu-NOC-973 catalysts was evaluated in the catalytic hydrochlorination of five terminal alkynes (propyne, propargyl bromide, 1-butyne, 1-pentyne and phenylacetylene) and two internal alkynes (2-butyne, phenylpropyne), as shown in Figure 4i and Table S6, several relevant reactions in the production of functional materials and synthesis of intermediate for medicine.¹⁸ The results indicate that the Cu-NOC-973 catalyst exhibits high conversion rates (>75%)*

for seven compounds containing alkynyl, with selectivity for the reaction products exceeding 90%. Furthermore, the large-scale reactions and catalyst recyclability studies indicated that the Cu-NOC-973 catalyst furnished high stability, with the conversion and selectivity exceeding 90% after five successive cycles (Figure S26). Therefore, the construction of the synergistic integrated catalytic pairs system demonstrates considerable versatility in enhancing the conversion efficiency of alkynes and the selective transformation of target products.”

7. Minor corrections:

Define CuPc at first use.

Line 156 Spelling error “Rede”

Figure 3e and 3f. x-axis should be “Radial distance” instead of radical.

Correct valance to valence throughout manuscript

Reply: We again apologize for our carelessness. We have carefully checked the whole manuscript and corrected the above errors in the revised manuscript.

Again, we thank you for your careful readings, positive comments, and valuable suggestions.

We attached a copy of the revised manuscript at the end of this document. This copy is the same as the revised manuscript, except that some parts are colored to make the revisions notable.

We hope the Reviewers and the Editors will be satisfied with the revisions for the original manuscript.

Thanks and Best regards!

Yours Sincerely,

Feng Ryan Wang, Xiaonian Li and Jia Zhao